# WestWorld: A Knowledge-Encoded Scalable Trajectory World Model for Diverse Robotic Systems

Yuchen Wang [1] [*]   Jiangtao Kong [1] [*]   Sizhe Wei [2]   Xiaochang Li [1]   Haohong Lin [3]   Hongjue Zhao [4]   Tianyi Zhou [5]
Lu Gan [2]   Huajie Shao [1]

## Abstract

Trajectory world models play a crucial role in robotic dynamics learning, planning, and control. While recent works have explored trajectory world models for diverse robotic systems, they struggle to scale to a large number of distinct system dynamics and overlook domain knowledge of physical structures. To address these limitations, we introduce `WestWorld`, a knoWledge-Encoded Scalable Trajectory World model for diverse robotic systems. To tackle the scalability challenge, we propose a novel system-aware Mixture-of-Experts (Sys-MoE) that dynamically combines and routes specialized experts for different robotic systems via a learnable system embedding. To further enhance zero-shot generalization, we incorporate domain knowledge of robot physical structures by introducing a structural embedding that aligns trajectory representations with morphological information. After pretraining on 89 complex environments spanning diverse morphologies across both simulation and real-world settings, `WestWorld` achieves significant improvements over competitive baselines in zero- and few-shot trajectory prediction. Additionally, it shows strong scalability across a wide range of robotic environments and significantly improves performance on downstream model-based control for different robots. Finally, we deploy our model on a real-world Unitree Go1, where it demonstrates stable locomotion performance. The code is available at https://github.com/511205787/WestWorld.

## 1. Introduction

Trajectory world models (Ha & Schmidhuber, 2018; Wang et al., 2025; Yin et al., 2025) are essential for robotic dynamics learning (Xie et al., 2025), planning, and control based on *low-level sensory data*. However, building a trajectory world model for diverse robotic systems poses two key challenges: *i) sensor and actuator heterogeneity*, where the variance in types and sampling rates hinders shared representations, and *ii) system dynamics gaps* caused by diverse kinematic structures across different robotic systems.

To address these challenges, a few recent studies (Schubert et al., 2023; Yin et al., 2025) discretize continuous states and actions across diverse systems into tokens via quantization and leverage flexible Transformer architectures for joint training. Although these approaches enable multi-system pretraining within a single dense model, they still face *scalability and generalization limitations* across diverse robotic dynamics for two main reasons. *First*, existing approaches force different system dynamics to share a common set of model parameters, leading to gradient conflicts and negative transfer that impede effective scaling as robot diversity grows. *Second*, these methods overlook robot morphological information when modeling trajectories, thereby lacking the physical inductive biases required for zero-shot generalization to unseen robotic systems.

To overcome these limitations, we develop `WestWorld`, a knowledge-encoded scalable trajectory world model that incorporates domain knowledge of robot morphology to learn the underlying dynamics of diverse robotic systems (see Fig. 1). Developing such a model poses two *key challenges*: *i)* learning distinct system dynamics at scale while avoiding task interference across different robots, and *ii)* incorporating physical structural information as an inductive bias to enhance zero-shot generalization. To address the first challenge of scalability, we propose a system-aware mixture-of-experts (Sys-MoE) that implicitly learns distinct system dynamics through expert learning. Unlike existing trajectory world models, which learn multiple system dynamics using a single large dense model, the proposed Sys-MoE dynamically combines and routes specialized experts for different robotic systems via a learnable system embedding.

[1]William & Mary [2]Georgia Institute of Technology [3]Carnegie Mellon University [4]University of Illinois at Urbana-Champaign [5]Mohamed bin Zayed University of Artificial Intelligence. Correspondence to: Huajie Shao <hshao@wm.edu>.

*Proceedings of the 43rd International Conference on Machine Learning*, Seoul, South Korea. PMLR 306, 2026. Copyright 2026 by the author(s).

This design mitigates task interference across robots, thus significantly improving scalability. For the second challenge of generalization, we introduce a structure-based channel embedding that aligns low-level state trajectories with morphology information, thereby improving the model's ability to generalize to unseen robotic systems.

We pretrain the proposed `WestWorld` on 89 complex environments using a combination of simulated and real-world data. Extensive experiments show that our method substantially outperforms strong baselines in both zero-shot and few-shot trajectory prediction. Moreover, it enables scalable training across diverse robotic systems without sacrificing performance and significantly improves the performance of downstream tasks such as model-based control.

**Our contributions include:** 1) We propose `WestWorld`, a novel system-aware MoE architecture for scaling up the training of trajectory world models across diverse robotic systems; 2) We introduce knowledge-encoded structural embedding that provides an explicit inductive bias to enhance zero- and few-shot generalization to unseen robotic systems; 3) We conduct extensive experiments to verify the scalability and generalizability of our method, showing its superiority over strong baselines; and 4) We apply our model to downstream model-based control and further extend it to real-world Unitree Go1 deployment, demonstrating its strong performance in planning tasks.

## 2. Related Work

**World Models for Single Robots.** World models (Ha & Schmidhuber, 2018) serve as a foundational tool for sequential decision-making in embodied agents (Wei et al., 2025; Long et al., 2025), enabling them to model, understand, and predict environmental dynamics. In robotics, one line of work (Agarwal et al., 2025; Chi et al., 2025) leverages video generation models as world models to synthesize temporally coherent observations, thereby implicitly capturing underlying physical dynamics. Another line of research develops action-conditioned *dynamics* world models that explicitly predict future states for planning and control (Guo et al., 2025; Ebert et al., 2018; Zhu et al., 2025; Hansen et al., 2022; Chua et al., 2018). These dynamics world models can be broadly categorized into video world models and trajectory world models, offering complementary perspectives for learning physical dynamics.

In this work, we focus on *studying low-level trajectory world models*. Generalizing such models across diverse robotic systems is non-trivial, since trajectories are derived from heterogeneous sensors and actuators with mismatched channel semantics across systems. Thus, most existing trajectory world models (Wu et al., 2023; Hansen et al., 2022; Chua et al., 2018) are tailored to a single robot, and transferring to new platforms typically requires retraining or substantial adaptation. In contrast, our work aims to learn a unified trajectory world model that scales across diverse robotic systems.

**Trajectory World Model for Diverse Robotics.** Recent works (Hansen et al., 2024; Yin et al., 2025; Schubert et al., 2023) have explored trajectory world models for diverse robotic systems. A key challenge is handling varying sensor and actuator dimensionalities across different robots. A common strategy is to zero-pad inputs to a shared maximum dimension (Hansen et al., 2024). However, padding-based approaches suffer from dimensionality limits and often degrade generalization across environments (Yin et al., 2025). To address this, a few recent studies (Schubert et al., 2023; Yin et al., 2025) treat states and actions as token sequences and jointly train flexible Transformer architectures across multiple robots. Despite enabling joint pretraining across diverse robots, their scalability and generalization remain limited. First, most existing methods learn heterogeneous robotic systems using a single shared set of model parameters, which induces cross-system interference as robot diversity grows and makes scaling difficult. Second, these methods treat trajectories purely as token sequences while ignoring robot morphology information, which results in poor zero-shot generalization to unseen robotics.

Unlike prior works, we propose a system-aware MoE model that incorporates robot structural information to enable both scalable pretraining and strong zero-shot performance.

## 3. Preliminaries

**Notations**. The detailed descriptions of important notations are presented in Table 5 in Appendix A.

**Problem Statement**. A robotic system can be viewed as a controlled dynamical process with state space $\mathcal{S}$, action space $\mathcal{A}$, and transition dynamics $\boldsymbol{f}$. In practice, the state $\boldsymbol{s}_t \in \mathcal{S}$ consists of physical sensor readings such as joint positions and joint velocities, while the action $\boldsymbol{a}_t \in \mathcal{A}$ represents control commands such as joint torques. Model-based control relies on an internal dynamics model to support planning by rolling out candidate action sequences in imagination (Long et al., 2025).

In this work, we use a *trajectory world model* purely as a dynamics model (Sekar et al., 2020; Long et al., 2025; Ha & Schmidhuber) that predicts future states $\boldsymbol{s}$ conditioned on action interactions $\boldsymbol{a}$. Formally, a world model parameterizes a transition distribution

$$p_\theta(\boldsymbol{s}_{t+1} \mid \boldsymbol{s}_{1:t}, \boldsymbol{a}_{1:t}), \qquad \boldsymbol{s}_t \in \mathcal{S}, \ \boldsymbol{a}_t \in \mathcal{A}, \qquad (1)$$

which can be used to unroll state trajectories under a proposed action sequence. A rollout induces a trajectory

$$\tau = (\boldsymbol{s}_0, \boldsymbol{a}_0, \boldsymbol{s}_1, \boldsymbol{a}_1, \ldots, \boldsymbol{s}_t, \boldsymbol{a}_t), \qquad (2)$$

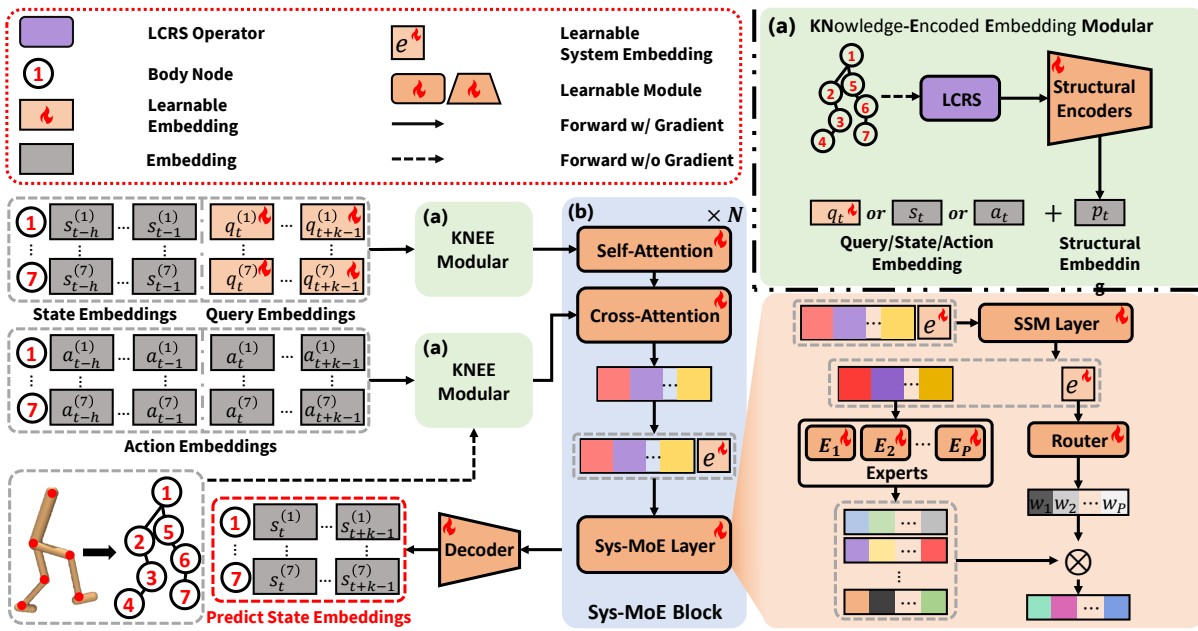

*Figure 1.* The overall architecture of our proposed `WestWorld`, consisting of two core components: (a) a Knowledge-Encoded Embedding Modular that injects structural embeddings as an inductive bias into trajectory representations, and (b) a System-aware MoE block that models diverse system dynamics via system-aware expert routing.

and the model is trained on the recorded trajectories $\mathcal{D} = \{\tau_i\}$ by maximizing predictive accuracy of future states.

*The goal of this work* is to develop a pretrained trajectory world model to learn the system dynamics across varying robotic systems and environments. Given a trajectory dataset from $n$ distinct robotic systems, $\{\mathcal{D}_1, \mathcal{D}_2, \ldots, \mathcal{D}_n\}$, our objective is to learn a single model $\theta$ that captures the dynamics of all $n$ systems. Specifically, given a history of $h$ past states and actions, the model tries to predict the next $k$ future states, conditioned on a sequence of $k$ future actions.

## 4. Proposed Method

### 4.1. Overview of **WestWorld**

To enable scalable pretraining and zero-shot generalization across diverse robotic systems, we propose `WestWorld`, a knowledge-encoded scalable trajectory world model with a system-aware Mixture-of-Experts (MoE) design. As shown in Fig. 1, the proposed model consists of two core components: (1) *Knowledge-Encoded Embedding Modular* and (2) *System-Aware MoE*.

The *core idea* is to first perform channel-wise normalization and discretize each scalar variable for tokenization. The resulting representations are then processed by *Knowledge-Encoded Embedding Modular*, which extracts the robot's morphological connectivity and injects structural embed-

dings as an inductive bias into trajectory representations. These structure-aware embeddings are subsequently fed into multiple *System-Aware MoE* blocks for dynamics modeling. Finally, a linear decoder maps the hidden states to future trajectory predictions. We detail these two core components in the following.

### 4.2. Knowledge-Encoded Embedding Modular

**Motivation.** Existing trajectory world models are predominantly data-driven, relying solely on state-action observations and largely ignoring domain knowledge that different robotic morphologies should obey distinct physical constraints. The lack of encoding explicit structural information makes it difficult for these models to capture the underlying system dynamics and limits their ability to generalize across environments. We hypothesize that robots with similar connectivity patterns often exhibit shared high-level dynamical behaviors (e.g., SLIP-like locomotion (Schwind, 1998)). This insight motivates us to incorporate morphological connectivity into model design as an inductive bias. Below, we first introduce the trajectory data tokenization before diving into proposed knowledge-encoded structural embedding.

**Trajectory Tokenization.** Given a trajectory, we treat each state or action dimension at time step $t$ as a *scalar channel*. Let $x_t^{(m)} \in \mathbb{R}$ denote the value of channel $m$ at time $t$, where $m$ represents the index of state channels or ac-

tion channels. We apply channel-wise min–max normalization, and discretize it into a $K$-bin categorical vector through $\phi : \mathbb{R} \to \mathbb{R}^K$ following (Yin et al., 2025). We further analyze the effect of different numbers of bins in Appendix D.2. We then map $\phi(x_t^{(m)})$ to a $d$-dimensional embedding via a learned projection. After that, we incorporate timestep embeddings, channel order index embeddings, and modality indicator (state or action) embeddings, yielding $z_t^{(m)} \in \mathbb{R}^d$. For convenience, we stack the per-channel embeddings from states and actions at time step $t$ into $\boldsymbol{S}_t \triangleq [\boldsymbol{s}_t^{(1)}, \ldots, \boldsymbol{s}_t^{(M_s)}]^\top \in \mathbb{R}^{M_s \times d}$ and $\boldsymbol{A}_t \triangleq [\boldsymbol{a}_t^{(1)}, \ldots, \boldsymbol{a}_t^{(M_a)}]^\top \in \mathbb{R}^{M_a \times d}$, where $M_s$ and $M_a$ are the numbers of state and action channels.

**Knowledge-Encoded Structural Embedding.** To incorporate morphology structure priors into latent representations, we introduce a knowledge-encoded structural embedding, as shown in Fig. 1 (a). Specifically, we first model each articulated object as a rooted kinematic tree and convert it to a binary tree using the left-child-right-sibling (LCRS) transformation (Hong et al., 2021). Each body node is assigned three traversal indices from pre-/in-/post-order walks. For object $i$ and its body node $j$, let $\left(\pi_{\text{pre}}^{i,j}, \pi_{\text{in}}^{i,j}, \pi_{\text{post}}^{i,j}\right)$ denote these indices. In scenes with multiple articulated objects, we additionally assign an object identifier $\pi_{\text{obj}}^i$: the robot is indexed as $\pi_{\text{obj}}^i = 0$, and other objects are ordered by increasing Euclidean distance to the robot (see Appendix B for an example). With this tuple indices, we can uniquely identify each robot body node in the LCRS-converted binary tree derived from the robot's structure. Then, we embed these indices to obtain a structure embedding:

$$
\begin{aligned}
\boldsymbol{p}^{(i,j)} = \text{Concat}\Big( &\boldsymbol{e}_{\text{obj}}\big(\pi_{\text{obj}}^i\big), \ \boldsymbol{e}_{\text{pre}}\big(\pi_{\text{pre}}^{i,j}\big), \\
&\boldsymbol{e}_{\text{in}}\big(\pi_{\text{in}}^{i,j}\big), \ \boldsymbol{e}_{\text{post}}\big(\pi_{\text{post}}^{i,j}\big)\Big).
\end{aligned} \tag{3}
$$

where each $\boldsymbol{e}_{\{\text{obj,pre,in,post}\}}(\cdot)$ denotes a structural encoder that maps a discrete index to a $d/4$-dimensional vector, and their concatenation forms $\boldsymbol{p}^{(i,j)} \in \mathbb{R}^d$. Finally, we inject morphology knowledge by adding $\boldsymbol{p}^{(i,j)}$ to the corresponding state/action embeddings, yielding structure-aware trajectory embeddings that are used as inputs to our model.

### 4.3. System-Aware MoE Block

**Motivation.** Robotic systems with diverse morphologies often exhibit markedly different dynamics, making it difficult to develop a single unified model that accurately captures their underlying dynamics. When such dissimilar dynamics are trained simultaneously using shared parameters, optimization is prone to gradient conflicts and task interference, leading to poor scalability. To address this challenge, we introduce a novel system-aware Mixture-of-Experts (Sys-MoE) block for learning distinct system dynamics, in which each expert tries to learn part of underlying system dynam-

ics. Our *key insight* is that complex system dynamics can be effectively approximated by composing a set of basis dynamics with system-dependent coefficients.

**Block design.** To scale joint training across diverse robotic systems while mitigating interference, we parameterize the transition model in Eq. (1) with a stack of *Sys-MoE Blocks*. Each block contains two parts: *i)* an attention-based aggregation module that fuses state–action information, and *ii)* a system-aware MoE layer that captures diverse system dynamics. We detail the two parts below.

*i)* **Attention-based aggregation.** As shown in Fig. 1(b), we use attention to aggregate information across state channels and to inject action-dependent control signals, while naturally supporting variable state/action dimensionalities across systems. Concretely, we apply: 1) self-attention to capture correlations among state variables, and then use 2) cross-attention to condition state features on the action embeddings. To enable $k$-step prediction in a single forward pass, we concatenate the history state embeddings with $k$ learnable query embeddings $\{\boldsymbol{q}_t, \ldots, \boldsymbol{q}_{t+k-1}\}$, which serve as latent queries for future states.

At each time step, self-attention is computed as

$$
\tilde{\boldsymbol{S}}_t = \text{LN}\Big(\boldsymbol{S}_t + \text{Self-Atten}(\boldsymbol{S}_t)\Big), \tag{4}
$$

where self-attention is applied along the state channel, and $\text{LN}(\cdot)$ is layer normalization. We then condition $\tilde{\boldsymbol{S}}_t$ on the action embeddings $\boldsymbol{A}_t$ via multi-head cross-attention:

$$
\hat{\boldsymbol{S}}_t = \text{LN}\Big(\tilde{\boldsymbol{S}}_t + \text{Cross-Atten}(\tilde{\boldsymbol{S}}_t, \boldsymbol{A}_t)\Big), \tag{5}
$$

This operation injects action-dependent signals while remaining compatible with variable action dimensionalities.

*ii)* **System-aware MoE layer.** After obtaining the action-conditioned latent states above, we model continuous-time system dynamics using a system-aware Mixture-of-Experts (Sys-MoE) layer, as shown in Fig. 1(b). Unlike the MoE design commonly used in large language models, where routing selects experts to directly produce token embeddings from the input, our routing is *system-aware*. Specifically, we introduce a learnable system embedding that propagates through the SSM to extract system-level properties of the underlying dynamics. The resulting system embeddings are then used to compute mixture weights over experts, so that the model forms a system-conditioned combination of different experts.

Let $L = h + k$ be the number of state embeddings after concatenating history states with $k$ queries. For each state channel $m$, we denote the attention outputs as $\hat{\boldsymbol{S}}_{1:L}^{(m)} = \{\hat{\boldsymbol{s}}_{t-h:t-1}^{(m)}, \hat{\boldsymbol{s}}_{t:t+k-1}^{(m)}\}$, where the last $k$ tokens correspond to the query positions. We append a learnable system embed-

ding $e \in \mathbb{R}^d$ to attention outputs, yielding

$$\overline{S}^{(m)} = \text{Concat}\left(\hat{S}^{(m)}_{1:L}, e\right). \qquad (6)$$

We apply an SSM layer to obtain outputs $U_\ell$:

$$U^{(m)}_{1:L+1} = \text{SSM}(\overline{S}^{(m)}), \qquad (7)$$

where $U^{(m)}_{1:L+1}$ denotes the SSM outputs for all embeddings. In our implementation, $\text{SSM}(\cdot)$ follows a Mamba-style selective SSM (Gu & Dao, 2024), enabling causal computation and efficient long-range dependency modeling.

We use the output of the system embedding, $U_{L+1}$, to extract system-aware properties for routing. A router produces mixture weights over $P$ experts via a softmax gate:

$$w = \text{Softmax}(\text{Router}(U_{L+1})) \in \mathbb{R}^P. \qquad (8)$$

Given the output $U^{(m)}_{1:L}$, the Sys-MoE block outputs $Y^{(m)}_{1:L}$ is computed as a weighted combination of expert predictions:

$$Y^{(m)}_{1:L} = \sum_{p=1}^{P} w_p\, E_p(U^{(m)}_{1:L}), \qquad (9)$$

where $w_p$ is the $p$-th entry of $w$, and each expert $E_p(\cdot)$ is implemented as an MLP. Finally, we stack multiple Sys-MoE blocks to increase expressivity for complex system dynamics.

### 4.4. Objective Function.

After stacking *Sys-MoE Blocks*, we obtain the per-channel output sequence $Y^{(m)}_{1:L}$ We apply a linear decoder head to produce logits over $K$ uniform bins. Concretely, for each channel $m$ outputs, we compute

$$P^{(m)}_L = \text{Softmax}(W_{\text{dec}}\, Y^{(m)}_{1:L}) \in [0,1]^K. \qquad (10)$$

We train the model with a next-token cross-entropy loss on the state channels to match the categorical representation of the inputs:

$$\mathcal{L}_{\text{CE}} = -\sum_{\ell=1}^{L-1} \sum_{m=1}^{M_s} \phi(x^{(m)}_{\ell+1})^\top \log P^{(m)}_\ell. \qquad (11)$$

During inference time, we run the model in a *sequence-to-sequence manner*, enabling multi-step prediction in a single forward pass.

## 5. Experiments

In this section, we pretrain `WestWorld` on large-scale, diverse robotic datasets and conduct extensive experiments to evaluate: (i) zero-shot generalization to unseen environments, (ii) few-shot adaptation under domain shifts, (iii) scalability as the number of pretraining environments increases, and (iv) improvements in downstream control tasks across diverse robotic systems enabled by pretraining.

*Table 1.* Zero-shot generalization performance of different models on three dynamical systems. Errors are computed in the normalized space and reported as MAE and MSE ($\times 10^{-2}$); lower is better.

| Method | Walker2d ($\times 10^{-2}$) | | Hopper ($\times 10^{-2}$) | | Franka ($\times 10^{-2}$) | |
|---|---|---|---|---|---|---|
| | MAE ↓ | MSE ↓ | MAE ↓ | MSE ↓ | MAE ↓ | MSE ↓ |
| MLPEnsemble | 26.006 | 12.028 | 19.987 | 7.216 | 12.164 | 4.271 |
| TDM | 20.122 | 6.428 | 17.634 | 5.076 | 23.686 | 8.435 |
| Trajworld | 22.261 | 8.623 | 17.388 | 5.441 | 13.102 | 5.127 |
| Ours | **16.350** | **5.064** | **13.731** | **3.368** | **7.737** | **2.539** |

**Diverse Pretraining Datasets.** For pretraining the proposed trajectory world model, we collect a large amount of simulated and real-world data: i) UniTraj dataset (Yin et al., 2025), which contains 80 simulated robotic environments; and ii) 9 real-world robot-arm datasets from the Open X-Embodiment (Vuong et al., 2023). A detailed list of the pretraining and evaluation environments used in our experiments is provided in Appendix C.

**Baseline Methods.** We compare our method against several state-of-the-art trajectory world models. (1) *MLP Ensemble* (Chua et al., 2018): a widely used baseline in model-based RL for learning probabilistic dynamics through an ensemble of multilayer perceptrons. (2) *TDM* (Schubert et al., 2023): a Transformer-based model built upon the Gato architecture, which flattens spatial and temporal features into a single sequence and applies one-dimensional attention for autoregressive prediction. (3) *TrajWorld* (Yin et al., 2025): a Transformer-based trajectory model that employs temporal-variate attention for autoregressive rollout.

**Implementation Details.** We follow each baseline's original pretraining configuration. Detailed training and implementation settings for `WestWorld` and all baselines are provided in Appendix D. For fairness, all baseline models are pretrained from scratch on the above same dataset as the proposed `WestWorld`.

### 5.1. Main Results

**Evaluation on Zero-shot Performance**. We first evaluate the zero-shot performance of our model on three unseen robotic environments that share similar structural morphology with those in the pretraining data. Specifically, we use datasets from three environments as our testbeds. These include *Hopper* and *Walker2D* from D4RL (Fu et al., 2020), as well as a *real-world* dataset of a mobile *Franka* manipulator interacting with articulated objects (Schiavi et al., 2023). We evaluate 100-step consecutive predictions using a 50-step history window as input. Mean Absolute Error (MAE) and Mean Squared Error (MSE) are used to assess the accuracy of long-horizon prediction.

As shown in Table 1, our method achieves the best performance across all three unseen environments in long-horizon prediction. This improvement is attributed to the combina-

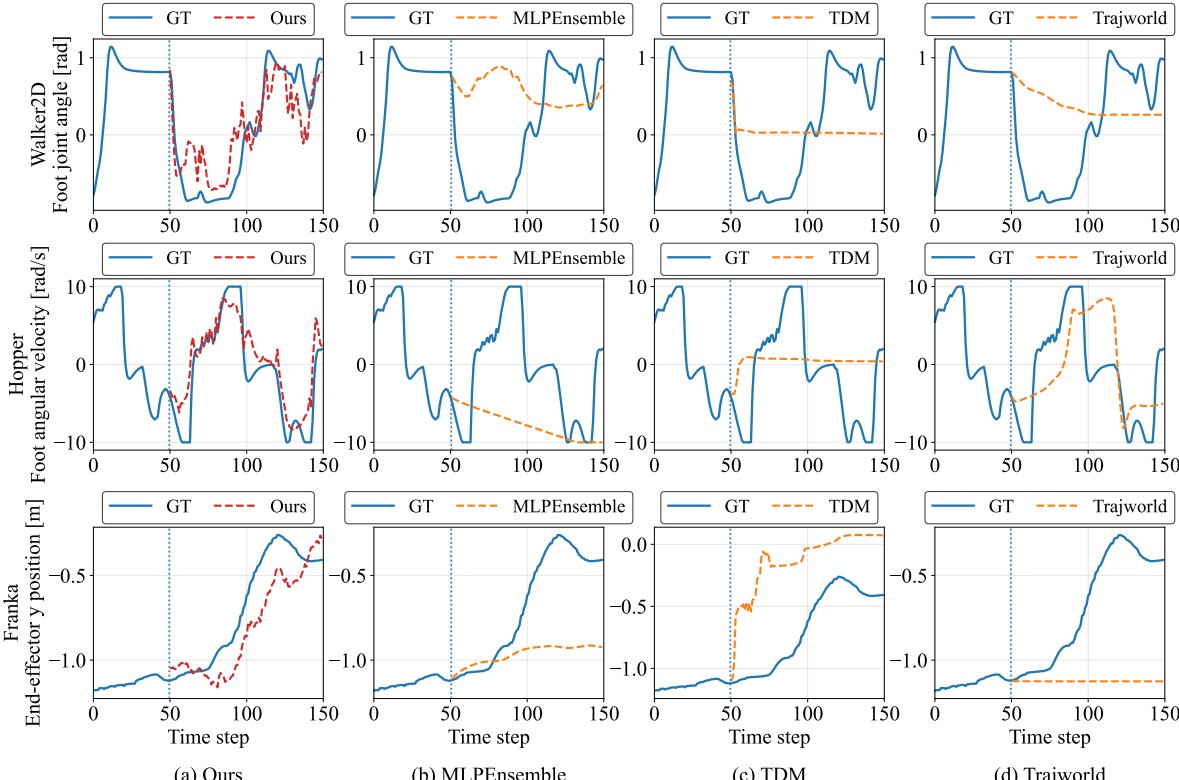

*Figure 2.* Trajectory plot comparison of our method and three baselines for 100-step rollout prediction on three robots: *Walker2D* foot joint angle, *Hopper* foot angular velocity, and *Franka* end-effector $y$ position, given a 50-step history window as input. We can observe that our method tracks the ground-truth dynamics substantially more closely than the baselines over the 100-step horizon.

*Table 2.* Few-shot generalization performance on three robotic systems. Results are computed in the normalized space and reported as MAE and MSE ($\times 10^{-2}$) averaged over three random seeds (mean ± standard deviation); lower is better.

| Method | Cassie ($\times 10^{-2}$) | | A1 ($\times 10^{-2}$) | | UR5 ($\times 10^{-2}$) | |
|---|---|---|---|---|---|---|
| | MAE ↓ | MSE ↓ | MAE ↓ | MSE ↓ | MAE ↓ | MSE ↓ |
| MLPEnsemble | $14.369 \pm 0.523$ | $4.416 \pm 0.368$ | $14.357 \pm 1.009$ | $3.707 \pm 0.523$ | $15.181 \pm 0.597$ | $4.936 \pm 0.322$ |
| TDM | $14.510 \pm 0.642$ | $3.404 \pm 0.300$ | $10.624 \pm 0.312$ | $2.151 \pm 0.109$ | $18.578 \pm 0.324$ | $5.510 \pm 0.175$ |
| Trajworld | $7.834 \pm 0.167$ | $1.697 \pm 0.109$ | $5.138 \pm 0.200$ | $0.900 \pm 0.050$ | $8.066 \pm 0.799$ | $2.117 \pm 0.433$ |
| Ours | $\mathbf{5.316 \pm 0.108}$ | $\mathbf{0.808 \pm 0.025}$ | $\mathbf{4.227 \pm 0.120}$ | $\mathbf{0.628 \pm 0.040}$ | $\mathbf{4.925 \pm 0.317}$ | $\mathbf{0.831 \pm 0.150}$ |

tion of our system-aware MoE architecture and the structural inductive bias introduced through morphology-aware design. The MoE design enables the model to learn distinct dynamics for different morphologies while mitigating task interference during pretraining. We further report unnormalized zero-shot errors in the original physical space in Appendix E.1, showing that the gains remain consistent under physically meaningful units. In addition, we visualize trajectory plots for all three robots in Fig. 2. We can see that `WestWorld` tracks the ground-truth dynamics substantially more closely than the baselines over the 100-step horizon. The reason is that, in a zero-shot setting on unseen but structurally similar systems, our model selects appropriate experts to produce accurate dynamics predictions, whereas baseline methods lack morphology-aware representations and fail to generalize.

**Evaluation on Few-shot Adaptation**. To examine the benefits of pretraining for learning distinct robotic dynamics under limited data, we also evaluate few-shot performance on three *real-world datasets* that exhibit a significant domain gap from the pretraining distribution: i) Cassie bipedal jumping (Acosta et al., 2022), ii) Unitree A1 quadruped locomotion (Tang et al.), and iii) UR5 tabletop manipulation (Zhou et al., 2023). For each dataset, we fine-tune using only 10 episodes, employ early stopping based on performance on a validation split, and report MAE and MSE on held-out test trajectories.

We can see from Table 2 that our method consistently outperforms all baselines across the three robotic systems despite

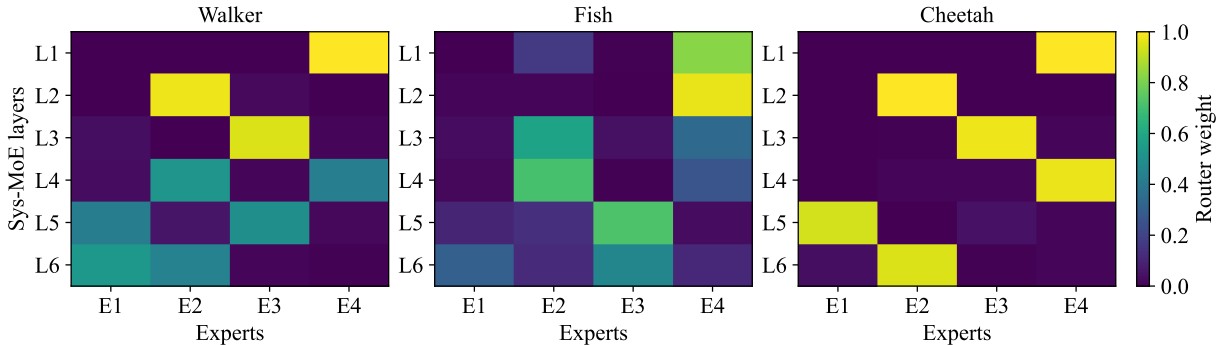

*Figure 3.* Sys-MoE routing weights across six layers (L1–L6), each containing four experts (E1–E4), for three robotic systems. Color indicates the router weight, where brighter values correspond to higher expert activation. The router exhibits near-sparse, system-dependent expert specialization, suggesting that different systems are modeled by different combinations of experts to capture their distinct dynamics.

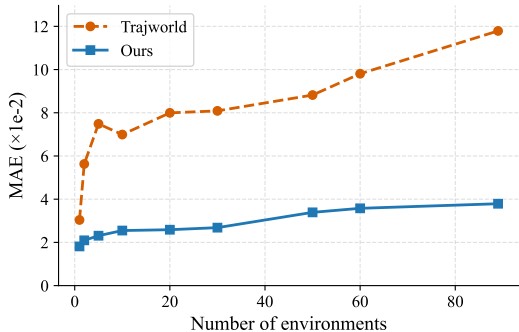

*Figure 4.* Comparison between our method against the best performing SOTA by scaling the number of environments.

the large morphology and dynamics gap from the pretraining data. This demonstrates that the pretrained model provides a strong initialization and improves performance even when adapting to systems with substantial domain differences.

To further quantify the impact of pretraining, we compare few-shot learning curves of `WestWorld` with and without pretraining. Overall, pretraining significantly improves final prediction accuracy across all three robots. Detailed results are provided in Appendix E.2.

**Evaluation on Scalability**. We further verify the scalability of our method by varying the $N$ number of robotic environments while keeping the data budget per environment fixed. Specifically, we evaluate $N \in \{1, 2, 5, 10, 20, 30, 50, 60, 89\}$ environments. Due to the different data availability across the expanded settings, the exact training and evaluation splits vary slightly across $N$; detailed task-level subsets and split protocols are provided in Appendix C.4. We compare our method with the state-of-the-art `TrajWorld` under the same data split for each setting.

All models take a 50-step history window as input and produce 100-step consecutive predictions. Fig. 4 reports the long-horizon prediction errors at each $N$ environments. We observe that the accuracy of our method remains low and

does not vary significantly with increasing $N$. The results show that our method can simultaneously learn distinct system dynamics across diverse environments. Conversely, TrajWorld's performance degrades significantly as the number of environments increases. A plausible explanation is that optimizing a single shared model across multiple dissimilar dynamics exacerbates gradient interference and negative transfer, thereby limiting scalability (Chen et al., 2018).

To further analyze scalability, we visualize Sys-MoE routing weights for three distinct systems in Fig. 3. Across systems, the router exhibits sparse, system-dependent expert selection. These results support our key insight: complex dynamics can be effectively approximated by composing a set of basis dynamics modules with system-dependent coefficients. Such system-aware design mitigates interference in multi-system joint learning and enables scalable pretraining across diverse robotic systems.

**Evaluation on Downstream Control Task.** In addition, we evaluate whether pretraining improves downstream model-based control across diverse robotic systems, which aims to isolate the effect of pretraining on dynamics modeling. Jointly optimizing the controller or policy is a separate question and is not considered here. We consider three robotic systems with distinct dynamics: Walker2D, Hopper from OpenAI Gym (Towers et al., 2024), and Unitree Go1 (Alvarez-Padilla et al., 2025). For each system, we collect an offline trajectory dataset from the environment. We then compare two training regimes for each world model: *(i)* fine-tuning from a pretrained checkpoint and *(ii)* training from scratch under the same dataset. After training, we deploy the learned dynamics model within MPPI (Williams et al., 2015), a commonly used sampling-based MPC controller. Additional details of the MPPI implementation are provided in Appendix D.4.

We set the MPPI planning horizon to 100 for Walker2D and Hopper, and to 40 for Go1. The setting is challenging: MPPI relies on long-horizon rollouts, so small model errors

*Table 3.* Downstream model-based control performance using MPPI on Walker2D, Hopper, and Unitree Go1. We report accumulated episode reward (higher is better) averaged over the evaluation episodes. All methods are evaluated with fixed random seeds and identical MPPI hyperparameters for fair comparison.

| Method | Pretrain | Walker2d | Hopper | Go1 |
|---|---|---|---|---|
| | | Accumulate Reward ↑ | Accumulate Reward ↑ | Accumulate Reward ↑ |
| MLPEnsemble | ✗ | 119.18 | 147.37 | -1.07 |
| | ✓ | 190.51 | 200.56 | -0.31 |
| TDM | ✗ | 122.65 | 242.34 | -0.72 |
| | ✓ | 207.61 | 154.64 | 0.03 |
| Trajworld | ✗ | 395.23 | 366.26 | 0.05 |
| | ✓ | 1933.52 | 534.32 | 0.49 |
| Ours | ✗ | 707.61 | 554.92 | 0.43 |
| | ✓ | **2134.60** | **2253.51** | **2.20** |

*Table 4.* Ablation results under the zero-shot setting. We ablate the Sys-MoE layer by replacing it with a dense SSM, and ablate the structural encoding by removing it during pretraining. Results are computed in the normalized space and reported as MAE and MSE ($\times 10^{-2}$); lower is better.

| Method | Sys-MoE | Structural embedding | Walker2D ($\times 10^{-2}$) | | Hopper ($\times 10^{-2}$) | | Franka ($\times 10^{-2}$) | |
|---|---|---|---|---|---|---|---|---|
| | | | MAE ↓ | MSE ↓ | MAE ↓ | MSE ↓ | MAE ↓ | MSE ↓ |
| Ours w/o Sys-MoE | ✗ | ✓ | 18.707 | 6.797 | 15.978 | 4.630 | 9.392 | 2.873 |
| Ours w/o Structural embedding | ✓ | ✗ | 21.156 | 7.872 | 16.227 | 4.990 | 7.897 | 2.707 |
| Ours | ✓ | ✓ | **16.350** | **5.064** | **13.731** | **3.368** | **7.737** | **2.539** |

can compound and degrade control. In addition, the planner may explore actions that push the system outside the offline training distribution, further causing compounding errors and suboptimal control (Parthasarathy et al., 2025).

Table 3 reports the accumulated episode reward. We draw two key observations. First, for nearly all methods and systems, pretraining consistently improves control performance compared with training from scratch. This suggests that pretraining yields dynamics representations that generalize better under distribution shift between offline training and online MPC rollouts. Second, WestWorld achieves the best performance across all three systems under both training regimes, with particularly large gains after pretraining. This indicates that our scalable model design and morphology-informed inductive bias significantly improve downstream control performance.

Additionally, we conduct a **real-world deployment** on the Unitree Go1. For real-time execution, we distill WestWorld into a lightweight two-layer student model and fine-tune it using simulated Go1 control data. To provide a side-by-side comparison, we apply the same distillation, fine-tuning, and MPPI deployment protocol to the strongest baseline TrajWorld. In real-world deployment, the distilled WestWorld model successfully completes the straight-walking task toward the target goal (Fig. 5), while the distilled TrajWorld model fails to reliably stand up and walk forward. This result is consistent with the downstream control results in Table 3, where WestWorld achieves the best Go1 performance under the same MPPI

setting.

This setting is particularly challenging because MPPI relies on long-horizon rollouts, where small model errors can compound and degrade control performance. In addition, both models are trained and fine-tuned using simulation data, and sim-to-real gaps, including actuator and contact mismatch, ground friction variation, battery-dependent torque limits, and state-estimation noise, can further amplify rollout errors and lead to suboptimal control. Under this setting, the improved dynamics prediction of WestWorld enables more stable action selection in MPPI and transfers to real-world execution. Details of the distillation and deployment protocol are provided in Appendix G. A demo video is available at https://westworldrobot.github.io/.

## 5.2. Ablation Studies

We also explore the impact of two core components on model performance: 1) knowledge-encoded embedding (KNEE) modular and 2) system-aware Mixture-of-Experts (Sys-MoE) layer. Both ablation studies are performed during pretraining and subsequently evaluated under the same zero-shot experimental setting using three robotic systems: *Hopper*, *Walker2D*, and *Franka*. We report long-horizon dynamics prediction errors using MAE and MSE.

**Effect of the KNEE Modular.** To isolate the role of structural inductive bias, we remove the knowledge-encoded structural embedding during pretraining (denoted as "w/o structural embedding"). As shown in Table 4, removing structural embedding leads to a clear degradation on *Hop-*

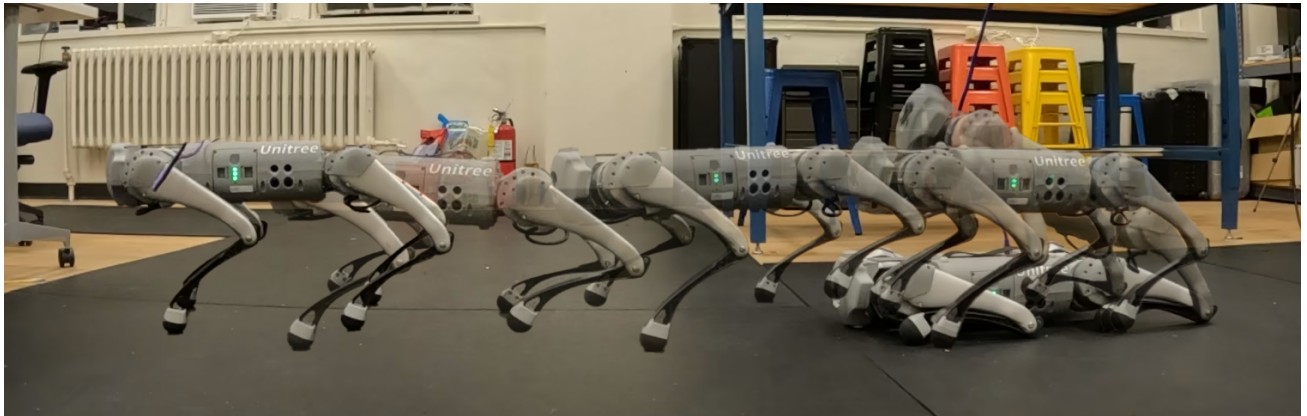

*Figure 5.* Real-world deployment on Unitree Go1. The distilled-and-fine-tuned `WestWorld` serves as the dynamics predictor in MPPI and enables the robot to walk straight toward the target goal. A side-by-side comparison with `TrajWorld` is provided in the project website at https://westworldrobot.github.io/.

*per* and *Walker2D*, which have more complex morphologies, while the drop on *Franka* is smaller. This results show that structural embedding is particularly beneficial for unseen complex robotic systems, where explicitly modeling physical connectivity helps align trajectory representations with system structure and improves zero-shot generalization.

**Effect of the Sys-MoE Layer.** To assess the importance of the Sys-MoE layer, we replace it with a dense SSM layer. For a fair comparison, we increase the depth of the dense-SSM variant so that its total number of parameters is comparable to that of our model. As shown in Table 4, this replacement degrades performance across tasks despite comparable model capacity. These results indicate that the Sys-MoE design is critical for jointly modeling diverse robotic dynamics, as it mitigates inter-task interference during multi-system training.

Based on the ablation study, we conclude that both KNEE and Sys-MoE are essential for model generalization and scalable pretraining.

### 5.3. Discussion

We also study parameter-efficient fine-tuning and provide a detailed inference-time latency comparison against autoregressive transformer baselines. Detailed analyses are deferred to Appendix F.

## 6. Conclusion and Limitation

In this work, we introduced `WestWorld`, a knowledge-encoded scalable trajectory world model designed for diverse robotics dynamics. Specifically, the proposed model leverages a Sys-MoE block to scale across diverse robotics dynamics and integrates morphology-aware structural embeddings to improve generalization ability. Extensive experimental results show that it significantly improves zero- and few-shot prediction performance on unseen robotic systems,

enhances model scalability, as well as boosts downstream model-based control performance.

Despite its remarkable performance, `WestWorld` currently focuses on trajectory modeling and does not explicitly incorporate visual observations. In the future, we will extend `WestWorld` into a multimodal world model that fuses both vision and trajectory signals.

## Acknowledgments

Research reported in this paper was sponsored in part by NSF CPS 2311086, NSF CIRC 716152, NSF RITEL 2506890, NAIRR 250288, and Faculty Research Grant at William & Mary 141446.

## Impact Statement

This paper aims to develop a general-purpose world model for diverse robotic systems to support prediction and control tasks. The proposed approach will significantly enhance the reliability and generalization of robotic systems operating in complex and dynamic environments. Beyond its technical contributions, this project is expected to catalyze interdisciplinary research at the intersection of machine learning, robotic systems, and control theory. Ultimately, it will facilitate the deployment of intelligent robotics in real-world applications, delivering broad benefits to both the research community and society.

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

# A. Notations

The table below summarizes the notation used in this paper. Lowercase letters (e.g., $x$) denote scalars, bold lowercase letters (e.g., $\boldsymbol{x}$) represent vectors, and bold uppercase letters (e.g., $\boldsymbol{A}, \boldsymbol{B}$) denote matrices.

*Table 5.* Summary of notations

| Notation | Definition |
|---|---|
| $\mathcal{S}$ | state space |
| $\mathcal{A}$ | action space |
| $\boldsymbol{s}$ | state |
| $\boldsymbol{a}$ | action |
| $n \in \mathbb{N}^+$ | number of robotics systems |
| $N$ | number of task-level environments in scalability evaluation |
| $\mathcal{D}_n$ | trajectory data for $n$-th robotics system |
| $h \in \mathbb{N}^+$ | history state time steps |
| $k \in \mathbb{N}^+$ | predicted future time steps |
| $m \in \mathbb{N}^+$ | channel index |
| $x_t^{(m)} \in \mathbb{R}$ | $m$-th channel states or actions at time $t$ |
| $\boldsymbol{\phi} : \mathbb{R} \to \mathbb{R}^K$ | discrete function |
| $\boldsymbol{z}_t^{(m)} \in \mathbb{R}^d$ | token embedding |
| $\boldsymbol{e}_{\text{time}} \in \mathbb{R}^d$ | timestep embedding |
| $\boldsymbol{e}_{\text{ch}} \in \mathbb{R}^d$ | channel order index embedding |
| $\boldsymbol{e}_{\text{type}} \in \mathbb{R}^d$ | modality embedding |
| $\boldsymbol{S}_t \in \mathbb{R}^{M_s \times d}$ | state embedding at time $t$, where $M_s$ denote the number of state channel |
| $\boldsymbol{A}_t \in \mathbb{R}^{M_a \times d}$ | action embedding at time $t$, , where $M_a$ denote the number of action channel |
| $\pi_{\text{obj}}^i \in \mathbb{N}^+$ | $i$-th object index |
| $\pi_{\text{pre}}^{i,j} \in \mathbb{N}^+$ | pre-order index for object $i$ and its body node $j$ |
| $\pi_{\text{in}}^{i,j} \in \mathbb{N}^+$ | in-order index for object $i$ and its body node $j$ |
| $\pi_{\text{post}}^{i,j} \in \mathbb{N}^+$ | post-order index for object $i$ and its body node $j$ |
| $\boldsymbol{p} \in \mathbb{R}^d$ | structural embedding |
| $\boldsymbol{q}_t \in \mathbb{R}^d$ | query embedding at time $t$ |
| $\tilde{\boldsymbol{S}}_t \in \mathbb{R}^{M_s \times d}$ | state hidden states after state self-attention at time $t$ |
| $\hat{\boldsymbol{S}}_t \in \mathbb{R}^{M_s \times d}$ | state hidden states after action-state cross-attention at time $t$ |
| $L \in \mathbb{N}^+$ | the length after concatenating history states with queries |
| $\overline{\boldsymbol{S}}^{(m)} = \left[\hat{\boldsymbol{S}}_{1:L}^{(m)}; e\right]$ | state embedding concate with system embedding |
| $\boldsymbol{e} \in \mathbb{R}^d$ | the learnable system embedding |
| $\boldsymbol{U}_\ell$ | $l$-th SSM output |
| $P \in \mathbb{N}^+$ | the number of expert |
| $\boldsymbol{Y}_{1:L}^{(m)} \in \mathbb{R}^{L \times d}$ | $m$-th channel System-Aware MoE Block output |
| $\boldsymbol{P}_\ell^{(m)} \in \mathbb{R}^K$ | categorical representation output |
| $M_s$ | the number of state channels |
| $M_a$ | the number of action channels |
| $K$ | the number of categorical bins |

# B. A Walk-Through Example for Structural Embedding

This section provides a concrete example of how we construct the knowledge-encoded structural embedding in Eq. (3). We use the Walker robot as an illustrative example.

**Step 1: Extract the robot kinematic tree.** Given the Walker structural file (e.g., an MJCF or URDF specification), we extract the articulated-body hierarchy and treat it as a rooted kinematic tree. Walker contains a single articulated object (the robot itself), so we set $\pi_{\text{obj}}^i = 0$. The robot has seven body nodes: torso, right thigh, right leg, right foot, left thigh, left leg, and left foot.

**Step 2: LCRS conversion and traversal ranks.** We convert the rooted kinematic tree into a binary tree via the left-child-right-sibling (LCRS) transformation (Hong et al., 2021). We then compute traversal ranks on the LCRS-converted binary tree, including pre-order, in-order, and post-order indices. For each body node $j$, we obtain a tuple of traversal ranks $\left(\pi_{\text{pre}}^{0,j}, \pi_{\text{in}}^{0,j}, \pi_{\text{post}}^{0,j}\right)$. For Walker, the indices for all seven body nodes are listed in Table 6.

*Table 6.* Walker example of LCRS traversal ranks used to construct structural embeddings. Here $\pi_{\text{obj}} = 0$ since the scene contains only the robot.

| Body | Global idx | $\pi_{\text{pre}}$ | $\pi_{\text{in}}$ | $\pi_{\text{post}}$ |
|---|---|---|---|---|
| torso | 0 | 0 | 6 | 6 |
| right_thigh | 1 | 1 | 3 | 5 |
| right_leg | 2 | 2 | 5 | 4 |
| right_foot | 3 | 3 | 4 | 3 |
| left_thigh | 4 | 4 | 2 | 2 |
| left_leg | 5 | 5 | 1 | 1 |
| left_foot | 6 | 6 | 0 | 0 |

**Step 3: Construct per-body structural embeddings.** For each body node $(i, j)$, we form the discrete index tuple

$$\left(\pi_{\text{obj}}^i, \pi_{\text{pre}}^{i,j}, \pi_{\text{in}}^{i,j}, \pi_{\text{post}}^{i,j}\right),$$

which uniquely identifies the node in the LCRS-converted binary tree (and also disambiguates multiple objects when present). We embed each index using a lookup table and concatenate the results as in Eq. (3):

$$\boldsymbol{p}^{(i,j)} = \text{Concat}\left(\boldsymbol{e}_{\text{obj}}(\pi_{\text{obj}}^i), \boldsymbol{e}_{\text{pre}}(\pi_{\text{pre}}^{i,j}), \boldsymbol{e}_{\text{in}}(\pi_{\text{in}}^{i,j}), \boldsymbol{e}_{\text{post}}(\pi_{\text{post}}^{i,j})\right) \in \mathbb{R}^d.$$

Finally, we inject the structural prior by adding $\boldsymbol{p}^{(i,j)}$ to the corresponding per-body token embedding (e.g., state/action/query tokens associated with body $j$). This yields structure-aware trajectory embeddings that are consistent across morphologies.

# C. Detailed Dataset Settings

In this appendix, we provide the detailed dataset settings in the experiments. This includes the datasets for pretraining, zero-shot evaluation, few-shot evaluation, scalability evaluation, and downstream model-based control.

## C.1. Pretraining Datasets

For pretraining the trajectory world model, we use both large-scale simulated and real-world data. Specifically, we use: (i) the UniTraj dataset (Yin et al., 2025), which aggregates trajectories from ExORL (Yarats et al., 2022), RL Unplugged (Gulcehre et al., 2020), JAT (Gallouédec et al., 2024), DB-1 (Wen et al., 2022), TD-MPC2 (Hansen et al., 2024), and Modular RL (Huang et al., 2020), covering in total 80 task-level simulated environments; and (ii) 9 real-world robot-arm datasets from Open X-Embodiment (Vuong et al., 2023). We first summarize the robot morphology categories in Table 7, and then list all 89 task-level environments used for pretraining in Table 8.

**Data preprocessing.** To stabilize training across heterogeneous robotic systems, we follow UniTraj (Yin et al., 2025) and apply channel-wise min–max normalization. For each robot and each feature dimension (e.g., joint position, joint velocity (angular or linear), or torque), we compute the empirical minimum and maximum values on the training split and rescale inputs to $[0, 1]$. Each trajectory segment is clipped or zero-padded to a maximum length of 150 time steps. Across all pretraining datasets, the maximum state dimension is 78 and the maximum action dimension is 21.

*Table 7.* A detailed list of diverse robotics used in the pretraining dataset. For robotics sharing the same name, we mark those from OpenAI Gym with an asterisk (*) and those from DeepMind Control Suite with a dagger (†).

| Dataset | Robot morphology categories |
| --- | --- |
| ExORL (Yarats et al., 2022) | Cartpole, Jaco, Quadruped, Walker† |
| RL Unplugged (Gulcehre et al., 2020) | Cartpole, Fish, Humanoid, Manipulator, Walker† |
| JAT (Gallouédec et al., 2024) | Double Pendulum*, Pendulum*, Pusher*, Reacher*, Swimmer* |
| DB-1 (Wen et al., 2022) | Acrobat, Ball In Cup, Cartpole, Cheetah-2-back, Cheetah-2-front, Cheetah-3-back, Cheetah-3-balanced, Cheetah-3-front, Cheetah-4-allback, Cheetah-4-allfront, Cheetah-4-back, Cheetah-4-front, Cheetah-5-back, Cheetah-5-balanced, Cheetah-5-front, Cheetah-6-back, Cheetah-6-front, Finger, Fish, Hopper†, Hopper-3, Hopper-5, Humanoid, Humanoid-2d-7-left-arm, Humanoid-2d-7-left-leg, Humanoid-2d-7-lower-arms, Humanoid-2d-7-right-arm, Humanoid-2d-7-right-leg, Humanoid-2d-8-left-knee, Humanoid-2d-8-right-knee, Humanoid-2d-9-full, Manipulator, Reacher, Swimmer6, Swimmer15, Walker†, Walker-2-flipped, Walker-2-main, Walker-3-flipped, Walker-3-main, Walker-4-flipped, Walker-4-main, Walker-5-flipped, Walker-5-main, Walker-6-flipped, Walker-6-main |
| TD-MPC2 (Hansen et al., 2024) | Acrobot, Ball In Cup, Cartpole, Cheetah†, Finger, Fish, Hopper†, Pendulum†, Reacher†, Walker† |
| Modular RL (Huang et al., 2020) | Cheetah-2-back, Cheetah-2-front, Cheetah-3-back, Cheetah-3-balanced, Cheetah-4-allback, Cheetah-4-back, Cheetah-4-front, Cheetah-5-back, Cheetah-5-balanced, Cheetah-5-front, Cheetah-6-back, Cheetah-6-front, Hopper-3, Hopper-5, Walker-2-flipped, Walker-3-flipped, Walker-4-flipped, Walker-5-flipped, Walker-6-flipped, Walker-7-flipped |
| Open X-Embodiment (Vuong et al., 2023) | Fanuc Mate 6-DoF industrial arm (Zhu et al., 2023), KUKA iiwa 7-DoF arm (Lee et al., 2019), Franka Emika Panda arm (Heo et al., 2025), Franka arm (Belkhale et al., 2023; Liu et al., 2025; Saxena et al., 2023; Sawhney et al., 2020; Chen et al., 2023), Sawyer arm (Gupta et al., 2022). |

## C.2. Zero-shot experiment datasets.

To evaluate zero-shot generalization, we consider three unseen environments: *Walker2D* and *Hopper* from OpenAI Gym (Towers et al., 2024), and a real-world dataset of a mobile *Franka* manipulator interacting with articulated objects (Schiavi et al., 2023). None of these systems appears during pretraining, but they share similar structural morphology with robots in the pretraining data, making them suitable for testing generalization to unseen environments.

For *Walker2D* and *Hopper*, we use expert demonstrations from D4RL (Fu et al., 2020) for evaluation. The *Walker2D* dataset

*Table 8.* Task-level environments used in the pretraining dataset.

| Component | Task-level environments | # |
|---|---|---|
| TD-MPC2 | `walker-stand, walker-walk, walker-run, cheetah-run, reacher-easy, reacher-hard, acrobot-swingup, pendulum-swingup, cartpole-balance, cartpole-balance-sparse, cartpole-swingup, cartpole-swingup-sparse, cup-catch, finger-spin, finger-turn-easy, finger-turn-hard, fish-swim, hopper-stand, hopper-hop, walker-walk-backwards, walker-run-backwards, cheetah-run-backwards, cheetah-run-front, cheetah-run-back, cheetah-jump, hopper-hop-backwards, reacher-three-easy, reacher-three-hard, cup-spin, pendulum-spin` | 30 |
| ExORL | `jaco-reach, quadruped-locomotion` | 2 |
| RL-Unplugged | `humanoid_run, manipulator-insert-ball, manipulator-insert-peg` | 3 |
| JAT | `inverted_double_pendulum, inverted_pendulum, pusher, reacher, swimmer` | 5 |
| DB-1 | `acrobot, cheetah_2_back, cheetah_2_front, cheetah_3_back, cheetah_3_balanced, cheetah_3_front, cheetah_4_allback, cheetah_4_allfront, cheetah_4_back, cheetah_4_front, cheetah_5_back, cheetah_5_balanced, cheetah_5_front, cheetah_6_back, cheetah_6_front, hopper_3, hopper_5, humanoid, humanoid_2d_7_left_arm, humanoid_2d_7_left_leg, humanoid_2d_7_lower_arms, humanoid_2d_7_right_arm, humanoid_2d_7_right_leg, humanoid_2d_8_left_knee, humanoid_2d_8_right_knee, humanoid_2d_9_full, manipulator-bring_ball, swimmer6, swimmer15, walker_2_flipped, walker_2_main, walker_3_flipped, walker_3_main, walker_4_flipped, walker_4_main, walker_5_flipped, walker_5_main, walker_6_flipped, walker_6_main` | 39 |
| Modular-RL | `walker_7_flipped` | 1 |
| Open X-Embodiment | `Fanuc-Mate-6DoF-tabletop-manipulation, KUKA-iiwa-7DoF-insertion, Franka-Emika-Panda-assemble-furniture, Franka-kitchen-manipulation, Franka-insertion, Franka-tabletop-manipulation, Franka-food-manipulation, Franka-play-with-object, Sawyer-pick-and-place` | 9 |
| **Total** | | **89** |

contains 1,267 episodes, and the *Hopper* dataset contains 1,029 episodes. The mobile *Franka* dataset contains 118 episodes.

### C.3. Few-shot experiment datasets.

To study few-shot adaptation under substantial domain shift, we evaluate on three real-world robotic datasets that are not covered by the pretraining distribution: (i) Cassie bipedal jumping (Acosta et al., 2022), (ii) Unitree A1 quadruped locomotion (Tang et al.), and (iii) UR5 tabletop manipulation (Zhou et al., 2023). All three datasets differ markedly from the simulated pretraining data in morphology and dynamics.

For each robot, we fine-tune `WestWorld` using only 10 training episodes, select the checkpoint via early stopping on a held-out validation split, and report MAE/MSE on a separate test split. UR5 provides 110 episodes in total, which we split into 10/50/50 for train/validation/test. Unitree A1 contains 20 episodes and is split into 10/5/5, and Cassie contains 28 episodes and is split into 10/6/12.

### C.4. Scaling of environmental datasets

In the environment-scaling study, we vary the number of pretraining environments and consider $N \in \{1, 2, 5, 10, 20, 30, 50, 60, 89\}$. For each value of $N$, we train on the task-level environment subset listed in Table 9. For $N \leq 50$, we use the constructed environment-scaling split, where each selected environment contains 1000 training episodes, 500 validation episodes, and 500 test episodes. For the $N = 60$ setting, due to data availability, each selected environment contains 500 training episodes, 250 validation episodes, and 250 test episodes. For the full $N = 89$ setting, we

train on the complete pretraining environment set and evaluate on the held-out validation split from the pretraining data. For each $N$, `WestWorld` and `TrajWorld` are trained and evaluated under the same split to ensure a fair comparison.

*Table 9.* Detailed task-level environment subsets and data split protocols for the environment-scaling study. For $N \leq 50$, each selected environment uses 1000/500/500 episodes for train/validation/test. For $N = 60$, each selected environment uses 500/250/250 episodes due to data availability. All compared methods use the same split for each $N$.

| $N$ | Selected task-level environments |
| --- | --- |
| 1 | Cartpole Swingup |
| 2 | Cartpole Swingup; Acrobot Swingup |
| 5 | Walker Walk Backwards; Finger Turn Easy; Cartpole Swingup; Reacher Three Easy; Cheetah Run Front |
| 10 | Walker Walk Backwards; Finger Turn Easy; Cartpole Swingup; Reacher Three Easy; Cheetah Run Front; Acrobot Swingup; Reacher Hard; Cup Catch; Cartpole Swingup Sparse; Finger Turn Hard |
| 20 | Walker Walk Backwards; Finger Turn Easy; Cartpole Swingup; Reacher Three Easy; Cheetah Run Front; Acrobot Swingup; Reacher Hard; Cup Catch; Cartpole Swingup Sparse; Finger Turn Hard; Cartpole Balance Sparse; Hopper Hop Backwards; Pendulum Spin; Cheetah Run Backwards; Fish Swim; Reacher Three Hard; Walker Walk; Finger Spin; Hopper Hop; Walker Run |
| 30 | Walker Walk Backwards; Finger Turn Easy; Cartpole Swingup; Reacher Three Easy; Cheetah Run Front; Acrobot Swingup; Reacher Hard; Cup Catch; Cartpole Swingup Sparse; Finger Turn Hard; Cartpole Balance Sparse; Hopper Hop Backwards; Pendulum Spin; Cheetah Run Backwards; Fish Swim; Reacher Three Hard; Walker Walk; Finger Spin; Hopper Hop; Walker Run; Hopper Stand; Cup Spin; Reacher Easy; Cheetah Jump; Pendulum Swingup; Cartpole Balance; Cheetah Run Back; Walker Stand; Cheetah Run; Walker Run Backwards |
| 50 | Cartpole Swingup; Walker Stand; Walker Walk; Walker Run; Quadruped Locomotion; Jaco Reach; Fish Swim; Cartpole Balance; Reacher Easy; Hopper Hop; Hopper Stand; Cup Catch; Acrobot Swingup; Cartpole Balance Sparse; Cartpole Swingup Sparse; Cheetah Jump; Cheetah Run; Cheetah Run Back; Cheetah Run Backwards; Cheetah Run Front; Finger Spin; Finger Turn Easy; Finger Turn Hard; Hopper Hop Backwards; Pendulum Swingup; Walker Run Backwards; Walker Walk Backwards; Reacher Hard; Swimmer; Inverted Double Pendulum; Pendulum Spin; Reacher Three Easy; Reacher Three Hard; Cup Spin; Inverted Pendulum; Furniture Bench; Humanoid Run; Manipulator Insert Ball; Manipulator Insert Peg; Pusher; Walker 2 Flipped; Hopper 5; Walker 4 Flipped; Hopper 3; Walker 7 Flipped; Walker 5 Flipped; Walker 3 Flipped; Walker 6 Flipped; Reacher; Stanford Mask-ViT |
| 60 | Cartpole Swingup; Walker Stand; Walker Walk; Walker Run; Quadruped Locomotion; Jaco Reach; Fish Swim; Cartpole Balance; Reacher Easy; Hopper Hop; Hopper Stand; Cup Catch; Acrobot Swingup; Cartpole Balance Sparse; Cartpole Swingup Sparse; Cheetah Jump; Cheetah Run; Cheetah Run Back; Cheetah Run Backwards; Cheetah Run Front; Finger Spin; Finger Turn Easy; Finger Turn Hard; Hopper Hop Backwards; Pendulum Swingup; Walker Run Backwards; Walker Walk Backwards; Reacher Hard; Swimmer; Inverted Double Pendulum; Pendulum Spin; Reacher Three Easy; Reacher Three Hard; Cup Spin; Inverted Pendulum; Furniture Bench; Humanoid Run; Manipulator Insert Ball; Manipulator Insert Peg; Pusher; Walker 2 Flipped; Hopper 5; Walker 4 Flipped; Hopper 3; Walker 7 Flipped; Walker 5 Flipped; Walker 3 Flipped; Cheetah 2 Back; Cheetah 2 Front; Walker 6 Flipped; Cheetah 3 Back; Cheetah 3 Balanced; Cheetah 4 Allback; Cheetah 4 Back; Cheetah 4 Front; Cheetah 5 Back; Cheetah 5 Balanced; Cheetah 5 Front; Cheetah 6 Back; Cheetah 6 Front |

## C.5. Downstream control task datasets.

In addition to trajectory prediction, we evaluate downstream model-based control using the learned `WestWorld`. We consider three robotic systems: Walker2D and Hopper from OpenAI Gymnasium (Towers et al., 2024), and the real-world Unitree Go1 locomotion dataset (Alvarez-Padilla et al., 2025). For each system, we construct an offline dataset for learning the dynamics model from MPPI rollouts and then evaluate control performance by deploying MPPI online.

**MPPI rollout collection.** We run MPPI for 100 episodes per system. For Walker2D and Hopper, each episode has 1000 environment steps, with planning horizon $H = 100$ and $N = 256$ sampled action sequences per step. For Go1, each episode has 2000 steps, with planning horizon $H = 40$ and $N = 30$ sampled action sequences per step. The MPPI cost definition for each task is provided in Appendix D.4.

**Training/validation data construction.** At each time step, we rank the $N$ sampled rollouts by MPPI cost and keep only the lowest-cost trajectories. Specifically, for training we retain the lowest 10% rollouts from the 100 MPPI episodes and use

them as supervised targets for dynamics learning. To improve data diversity, we additionally include all sampled rollouts from 3 extra episodes as augmentation. For validation, we use 5 episodes and retain the lowest 30% rollouts at each time step for early stopping and model selection. For testing, we report test performance by running MPPI online in the corresponding environment using the learned world model for rollouts.

## D. Detailed Experimental Settings

We present the detailed experimental settings in this Section. All pre-training experiments are conducted on a server equipped with 4 NVIDIA H200 GPUs, utilizing the PyTorch framework (Paszke et al., 2019).

### D.1. Hyperparameters for Models

This subsection summarizes the model hyperparameters used in our experiments. For all baseline methods, we follow the official implementations and the hyperparameter choices reported in prior work (Yin et al., 2025; Schubert et al., 2023).

**TrajWorld.** We use a TrajWorld architecture with discretized prediction using 256 uniform bins. The model has 6 Transformer blocks and 4 attention heads, with dropout rate 0.1. The hidden dimension is set to 256. Each block uses an MLP with hidden sizes $[1024, 256]$ and GeLU activation.

**TDM.** We use the TDM architecture with discretized prediction using 256 uniform bins. The model has hidden dimension 384, 6 Transformer blocks, and 4 attention heads, with dropout rate 0.1. Each block uses an MLP with hidden sizes $[1536, 384]$ and GeLU activation.

**MLPEnsemble.** For MLPEnsemble, we train an ensemble of transition models parameterized as a diagonal Gaussian over the next state, implemented with MLPs and optimized by negative log-likelihood using bootstrapped training samples. We use 7 MLPs, each with four hidden layers of width 640 (i.e., $[640, 640, 640, 640]$), and select the top 5 *elite* models by validation loss. To handle heterogeneous systems, we pad state and action vectors to fixed sizes before concatenation, producing a fixed-dimensional input to the MLP.

**WestWorld.** Our model follows the same discretized prediction setup as TrajWorld, using 256 uniform bins. The backbone consists of 6 Sys-MoE blocks with 4 attention heads, hidden dimension 256, and dropout rate 0.1. Each block uses an MLP with hidden size 512 and GeLU activation. For the SSM module in our Sys-MoE layer, implemented with a Mamba-style state-space model using state size 64, convolution width 4, expansion factor 2. Unless otherwise noted, each Sys-MoE layer contains $P = 4$ experts, which we select via validation as described below.

**Determining the number of experts.** Our Sys-MoE uses $P$ experts per Sys-MoE layer, where $P$ is a tunable hyperparameter. We select $P$ using a held-out validation split from the pretraining data. Specifically, we randomly sample 1% of trajectories from the 89 pretraining environments as a validation set and exclude them from optimization. This split is used only for model selection and does not overlap with any test environments. We select $P \in \{2, 4, 6, 8\}$ and report (i) the best validation MAE and (ii) the corresponding training step at which the best MAE is achieved (Table 10). While $P = 8$ achieves the lowest validation MAE, it requires substantially more training (about $1.6\times$ the steps of $P = 4$), leading to higher computational cost. The improvement from $P = 4$ to $P = 8$ is relatively modest on this validation split. Therefore, we use $P = 4$ as the default setting to balance accuracy and pretraining efficiency.

*Table 10.* Best validation MAE and the optimizer update step $s^\star$ at which the best MAE is achieved during pretraining. We report $s^\star$ in thousands of updates (k). MAE is computed in the normalized space. Lower values indicate better performance. † denotes the setting used in all experiments.

| #Experts $P$ | Val. MAE ↓ | Best step $s^\star$ (k updates) |
|:---:|:---:|:---:|
| 2 | 0.0409 | 108.6 |
| $4^\dagger$ | 0.0379 | 137.2 |
| 6 | 0.0390 | 176.7 |
| 8 | 0.0340 | 214.0 |

### D.2. Effect of the Number of Tokenization Bins

In this section, we quantitatively evaluate the effect of the number of discretization bins, using $K \in \{128, 256, 512\}$. We report the prediction errors on the Walker system in Table 11.

*Table 11.* Effect of the number of discretization bins on the Walker system. Errors are reported as MAE and MSE ($\times 10^{-2}$). Lower values are better.

| Number of bins | MAE $\downarrow$ | MSE $\downarrow$ |
|---|---|---|
| 128 | 1.455 | 0.098 |
| 256 | 1.415 | 0.095 |
| 512 | 1.404 | 0.094 |

Table 11 shows that increasing the number of bins from 128 to 256 improves prediction accuracy, whereas further increasing it to 512 yields only marginal gains. Considering the additional computational and memory costs introduced by larger output vocabularies, we use $K = 256$ as a practical trade-off between precision and efficiency in all main experiments.

### D.3. Training and Evaluation Settings

Below, we summarize the training and evaluation details for all experiments.

**Pretraining settings:** For all baseline methods, we follow the original pretraining protocols reported in prior work (Yin et al., 2025; Schubert et al., 2023). Specifically, for *TDM* and *TrajWorld*, we pretrain for a total of *up to* 1M gradient steps using the `Adam` optimizer with batch size 64 and learning rate $1 \times 10^{-4}$. We apply dropout with rate 0.05, weight decay $1 \times 10^{-5}$, and gradient clipping with norm 0.25. We use a warmup cosine decay learning-rate schedule with 10,000 warmup steps. For *MLPEnsemble*, we pretrain for *up to* 1M gradient steps with `Adam`, batch size 256, and learning rate $1 \times 10^{-4}$.

For our method, we pretrain for *up to* 1M gradient steps with `AdamW`, batch size 48, and learning rate $2 \times 10^{-4}$, while keeping dropout 0.05, weight decay $1 \times 10^{-5}$, gradient clipping norm 0.25, and the warmup cosine decay schedule with 10,000 warmup steps. Each training trajectory segment is clipped or zero-padded to a maximum length of 150 time steps. We use the first one-third of the segment (up to 50 steps) as the history window and predict the remaining future steps (up to 100 steps).

**Zero-shot evaluation settings:** We evaluate zero-shot long-horizon prediction on three *unseen* environments: *Walker2D* and *Hopper* from OpenAI Gym (Towers et al., 2024), and a real-world dataset of a mobile *Franka* manipulator interacting with articulated objects (Schiavi et al., 2023). For each dataset, we split each episode into trajectory segments of length 150. For all methods, we use the first 50 time steps as the history input and autoregressively roll out the next 100 time steps. We report MAE and MSE by comparing the 100-step predictions against the ground-truth future trajectory.

**Few-shot training settings:** For all methods in the 10-episode fine-tuning experiments, we use the `AdamW` optimizer with learning rate $1 \times 10^{-5}$ for up to 300 steps, with early stopping based on validation performance, and a maximum batch size of 12. Each training trajectory segment is clipped or zero-padded to a maximum length of 150 time steps. For the experiments that analyze pretraining benefits and quantify pretraining impact in Appendix E.2, we use `AdamW` with learning rate $1 \times 10^{-4}$ for up to 1500 steps for the train-from-scratch setting, again with validation-based early stopping and maximum batch size 12.

**Few-shot evaluation settings:** We evaluate few-shot long-horizon prediction on three real-world datasets: (i) Cassie bipedal jumping (Acosta et al., 2022), (ii) Unitree A1 quadruped locomotion (Tang et al.), and (iii) UR5 tabletop manipulation (Zhou et al., 2023). For Unitree A1 and UR5, all methods take the first 50 time steps as the history input and roll out the next 100 time steps. For Cassie, we use the same 50-step history window but predict only the next 50 time steps due to the shorter episode length. We report MAE and MSE by comparing the predictions against the corresponding ground-truth future trajectories.

**Downstream control training settings:** For all methods trained on the control datasets, we use the `AdamW` optimizer with learning rate $1 \times 10^{-5}$ for up to 50,000 steps, with early stopping based on validation performance, and a maximum batch size of 48. Each training trajectory segment is clipped or zero-padded to a maximum length of 100 time steps.

**Downstream control evaluation settings:** For control evaluation, we deploy each learned world model as the dynamics predictor within an MPPI controller. We set the prediction horizon to $H = 100$ for *Walker2D* and *Hopper*, and to $H = 40$ for *Unitree Go1*. At each control step, MPPI samples 256 candidate action sequences for *Walker2D/Hopper* and 30 for *Go1*, and uses the world model to roll out the corresponding trajectories for cost evaluation. All methods use identical MPPI

hyperparameters for fair comparison, as detailed in Appendix D.4.

### D.4. Description of Planning Algorithm

This section summarizes the trajectory optimization procedure used in our downstream control evaluations. Across Walker2D, Hopper, and Unitree Go1, we use a sampling-based MPC method, *Model Predictive Path Integral* (MPPI) (Williams et al., 2015), to optimize an open-loop action sequence over a finite horizon using the learned world model. At each control cycle, MPPI samples candidate action sequences, rolls them out through the world model, evaluates their trajectory costs, and updates the nominal action sequence via an exponentially weighted average. The first action is then executed in a receding-horizon manner.

**MPPI with a learned world model.** Let $f_\theta$ denote the learned dynamics model. Given the current state $s_0$, MPPI optimizes an action sequence $a_{0:H-1} = (a_0, \ldots, a_{H-1})$ over a horizon $H$ by rolling out

$$s_{t+1} = f_\theta(s_t, a_t), \qquad t = 0, \ldots, H - 1. \tag{12}$$

We maintain a nominal sequence $\mu_{0:H-1}$ and sample $N$ perturbed sequences:

$$a_t^{(n)} = \mu_t + \epsilon_t^{(n)}, \qquad \epsilon_t^{(n)} \sim \mathcal{N}(0, \Sigma), \qquad n = 1, \ldots, N, \ t = 0, \ldots, H - 1. \tag{13}$$

For each sample, we compute the trajectory cost

$$L^{(n)} = \sum_{t=0}^{H-1} \ell\left(s_t^{(n)}, a_t^{(n)}; s_t^{\text{ref}}\right), \tag{14}$$

and assign importance weights using a temperature parameter $\lambda$:

$$w^{(n)} \propto \exp\left(-\tfrac{1}{\lambda}\left(L^{(n)} - L_{\min}\right)\right), \qquad L_{\min} = \min_n L^{(n)}, \qquad \sum_{n=1}^{N} w^{(n)} = 1. \tag{15}$$

The nominal sequence is updated by the weighted average

$$\mu_t \leftarrow \sum_{n=1}^{N} w^{(n)} a_t^{(n)}, \qquad t = 0, \ldots, H - 1. \tag{16}$$

We execute the first action $\mu_0$ and repeat this procedure at the next control cycle.

For consistent reporting across tasks, we evaluate control performance using *accumulated reward* (higher is better), which is specified for each task below.

### D.4.1. WALKER2D

**Task.** We consider forward walking in the Walker2D environment (Towers et al., 2024). Let the Walker2D state at time $t$ be $s_t$ and the control input be $a_t$.

**Cost for forward walking.** We define the per-step cost as the negative of the standard Walker2D reward (i.e., lower cost corresponds to higher reward), following the Gymnasium formulation (Towers et al., 2024):

$$\ell(s_t, a_t) = -\left(r_{\text{healthy}}(t) + r_{\text{fwd}}(t) - c_{\text{ctrl}}(t)\right), \tag{17}$$

where $r_{\text{healthy}}(t)$ is a survival bonus when the walker remains healthy, $r_{\text{fwd}}(t)$ measures forward progress, and $c_{\text{ctrl}}(t)$ penalizes large control inputs.

Specifically, let $x_t$ denote the walker horizontal position (root $x$ coordinate) at time $t$, and let $\Delta t$ be the simulation time step. The forward progress reward is computed from the root velocity:

$$r_{\text{fwd}}(t) = w_{\text{fwd}} \frac{x_{t+1} - x_t}{\Delta t}, \tag{18}$$

with $w_{\text{fwd}} = 0.5$, and the control cost is

$$c_{\text{ctrl}}(t) = w_{\text{ctrl}} \|a_t\|_2^2, \tag{19}$$

with $w_{\mathrm{ctrl}} = 10^{-3}$. The survival bonus is

$$r_{\mathrm{healthy}}(t) = \begin{cases} 1, & \text{if the walker is healthy,} \\ 0, & \text{otherwise,} \end{cases} \tag{20}$$

where the health condition follows Gymnasium Walker2D and requires the state to remain within predefined ranges, including torso height $z \in [0.8, 2.0]$ and torso angle $\theta \in [-1.0, 1.0]$.

To improve stability of MPPI planning, we additionally include an expert-reference tracking term defined in the state space using an expert trajectory from D4RL (Fu et al., 2020):

$$\ell_{\mathrm{ref}}(t) = \left(\boldsymbol{s}_t - \boldsymbol{s}_t^{\mathrm{ref}}\right)^\top \boldsymbol{Q} \left(\boldsymbol{s}_t - \boldsymbol{s}_t^{\mathrm{ref}}\right), \tag{21}$$

where $\boldsymbol{s}_t^{\mathrm{ref}}$ is the reference state at time $t$ and $\boldsymbol{Q}$ is a diagonal weight vector. In our implementation, the reference is defined over the concatenated generalized positions and velocities $(\boldsymbol{q}, \dot{\boldsymbol{q}})$, and we use per-dimension weights

$$\boldsymbol{Q}_q = [0.0, 10.0, 10.0, 10.0, 1.0, 1.0, 10.0, 1.0, 1.0], \qquad \boldsymbol{Q}_{\dot{q}} = [0.01, 0.01, 0.01, 0.01, 0.01, 0.01, 0.01, 0.01, 0.01],$$

where the first position weight is set to 0 to ignore the root $x$ coordinate. We set $\boldsymbol{Q} = [\boldsymbol{Q}_q; \boldsymbol{Q}_{\dot{q}}]$.

Finally, the MPPI objective over horizon $H$ is

$$L_{\mathrm{walk}} = \sum_{t=0}^{H-1} \left(\ell(\boldsymbol{s}_t, \boldsymbol{a}_t) + \ell_{\mathrm{ref}}(t)\right). \tag{22}$$

**Accumulated reward.** We report the episode-level accumulated reward using the standard Walker2D reward:

$$R_{\mathrm{ep}} = \sum_{t=0}^{T-1} \left(r_{\mathrm{healthy}}(t) + r_{\mathrm{fwd}}(t) - c_{\mathrm{ctrl}}(t)\right). \tag{23}$$

**MPPI parameters.** We use temperature $\lambda = 0.25$, horizon $H = 100$, and $N = 256$ sampled trajectories per control cycle.

### D.4.2. HOPPER

**Task.** We consider forward hopping in the Hopper environment (Towers et al., 2024). Let the Hopper state at time $t$ be $\boldsymbol{s}_t$ and the control input be $\boldsymbol{a}_t$.

**Cost for forward hopping.** We define the per-step cost as the negative of the standard Hopper reward (i.e., lower cost corresponds to higher reward), following the Gymnasium formulation (Towers et al., 2024):

$$\ell(\boldsymbol{s}_t, \boldsymbol{a}_t) = -\left(r_{\mathrm{healthy}}(t) + r_{\mathrm{fwd}}(t) - c_{\mathrm{ctrl}}(t)\right), \tag{24}$$

where $r_{\mathrm{healthy}}(t)$ is a survival bonus when the hopper remains healthy, $r_{\mathrm{fwd}}(t)$ measures forward progress, and $c_{\mathrm{ctrl}}(t)$ penalizes large control inputs.

Specifically, let $x_t$ denote the hopper's horizontal position (root $x$ coordinate) at time $t$, and let $\Delta t$ be the simulation time step. The forward progress reward is

$$r_{\mathrm{fwd}}(t) = \frac{x_{t+1} - x_t}{\Delta t}, \tag{25}$$

and the control cost is

$$c_{\mathrm{ctrl}}(t) = w_{\mathrm{ctrl}} \|\boldsymbol{a}_t\|_2^2, \tag{26}$$

with $w_{\mathrm{ctrl}} = 10^{-3}$. The survival bonus is

$$r_{\mathrm{healthy}}(t) = \begin{cases} 1, & \text{if the hopper is healthy,} \\ 0, & \text{otherwise,} \end{cases} \tag{27}$$

where the health condition follows Gymnasium Hopper and requires the state to remain within predefined ranges (e.g., torso height and angle).

To improve stability of MPPI planning, we additionally include an expert-reference tracking term defined in the position and velocity space using a expert trajectory from D4RL (Fu et al., 2020):

$$\ell_{\mathrm{ref}}(t) = \left(\boldsymbol{s}_t - \boldsymbol{s}_t^{\mathrm{ref}}\right)^\top \boldsymbol{Q} \left(\boldsymbol{s}_t - \boldsymbol{s}_t^{\mathrm{ref}}\right), \tag{28}$$

where $s_t^{\text{ref}}$ is the reference state at time $t$, and $Q$ is a diagonal weight matrix with position and velocity weights. In our implementation, we set the position and velocity weights to $0.15$ and ignore the root $x$ position in the tracking loss.

Finally, the MPPI objective over horizon $H$ is

$$L_{\text{hop}} = \sum_{t=0}^{H-1} \Big( \ell(s_t, a_t) + \ell_{\text{ref}}(t) \Big). \tag{29}$$

**Accumulated reward.** We report the episode-level accumulated reward using the standard Hopper reward:

$$R_{\text{ep}} = \sum_{t=0}^{T-1} \Big( r_{\text{healthy}}(t) + r_{\text{fwd}}(t) - c_{\text{ctrl}}(t) \Big). \tag{30}$$

**MPPI parameters.** We use temperature $\lambda = 0.5$, horizon $H = 100$, and $N = 256$ sampled trajectories per control cycle.

### D.4.3. UNITREE GO1 ROBOT

**Task.** Following (Alvarez-Padilla et al., 2025), we consider straight walking on flat terrain. The robot tracks a sequence of planar waypoints while following an internal gait reference. Let the Go1 state at time $t$ be

$$s_t = \big[ p_t, \, q_t, \, q_t^j, \, v_t, \, \omega_t, \, \dot{q}_t^j \big],$$

where $p_t \in \mathbb{R}^3$ is the base position, $q_t \in \mathbb{H}$ is the base orientation quaternion, $q_t^j \in \mathbb{R}^{12}$ are joint angles, $v_t \in \mathbb{R}^3$ and $\omega_t \in \mathbb{R}^3$ are base linear/angular velocities, and $\dot{q}_t^j \in \mathbb{R}^{12}$ are joint velocities. The control input $a_t \in \mathbb{R}^{12}$ is the commanded joint action.

**Cost for straight walking.** Following (Alvarez-Padilla et al., 2025), we define a tracking objective that penalizes deviations from a desired state reference and a gait-based joint reference:

$$L_{\text{walk}} = \sum_{t=0}^{H-1} \Big[ \big( s_t^{\text{ref}} - s_t \big)^\top Q \big( s_t^{\text{ref}} - s_t \big) + \big( a_t^{\text{ref}} - a_t \big)^\top R \big( a_t^{\text{ref}} - a_t \big) \Big], \tag{31}$$

where $s_t$ and $a_t$ denote the robot state and control at time $t$, respectively. The state reference $s_t^{\text{ref}}$ includes the desired base position and attitude induced by the current waypoint. The control reference $a_t^{\text{ref}}$ is given by the gait scheduler (Raibert et al., 1989), and $Q$ and $R$ are diagonal weight matrices.

**Accumulated reward.** To evaluate performance on the *walk-straight* task, we follow the Gymnasium definition (Towers et al., 2024) and use the *forward_reward* for Go1. This reward measures the robot's planar displacement projected onto the instantaneous direction toward the current goal.

Let an episode be discretized into time steps $t = 0, 1, \ldots, T - 1$. Denote the robot base position in the world frame as $p_t \in \mathbb{R}^3$, and its planar component as $p_{t,xy} \in \mathbb{R}^2$. Let the current goal at time $t$ be $g_t \in \mathbb{R}^2$. For each simulation step, we record the planar position before advancing the dynamics, $p_{t,xy}^{\text{pre}}$, and after the dynamics step, $p_{t,xy}^{\text{post}}$:

$$p_{t,xy}^{\text{pre}} = p_{t,xy}, \qquad p_{t,xy}^{\text{post}} = p_{t+1,xy}. \tag{32}$$

We define the direction to the goal and its normalized form as

$$d_t = g_t - p_{t,xy}^{\text{pre}}, \qquad \hat{d}_t = \begin{cases} \dfrac{d_t}{\|d_t\|_2}, & \|d_t\|_2 \geq \varepsilon, \\ \mathbf{0}, & \text{otherwise,} \end{cases} \tag{33}$$

where $\varepsilon > 0$ is a small constant used to avoid numerical instability when $\|d_t\|_2$ is close to zero. Thus, $\hat{d}_t = \mathbf{0}$ only when the robot is sufficiently close to the goal (i.e., $\|d_t\|_2 < \varepsilon$); otherwise, $\hat{d}_t$ is the unit vector pointing from the current position to the goal. (we use $\varepsilon = 10^{-8}$). The planar displacement over one step is

$$\Delta p_t = p_{t,xy}^{\text{post}} - p_{t,xy}^{\text{pre}}. \tag{34}$$

The per-step goal-conditioned forward progress reward is then defined as

$$r_{\text{fwd}}(t) = \Delta p_t^\top \hat{d}_t. \tag{35}$$

Finally, we report the episode-level accumulated forward progress

$$R_{\text{fwd}} = \sum_{t=0}^{T-1} r_{\text{fwd}}(t). \tag{36}$$

**MPPI parameters.** We use temperature $\lambda = 0.1$, horizon $H = 40$, $N = 30$ sampled trajectories per control cycle.

## E. Additional Experiments

### E.1. Physical-space Zero-shot Evaluation

In the main experiments, we report MAE and MSE in the normalized space to ensure comparable evaluation across heterogeneous state dimensions and robotic systems. To provide a more physically interpretable evaluation, we additionally compute zero-shot prediction errors after mapping the predicted values back to their original physical units. Table 12 reports representative state dimensions from Walker2D, Hopper, and Franka, covering velocity, joint angle, position, and orientation quantities. The results show that WestWorld consistently achieves the lowest errors across all reported physical quantities, indicating that the improvements observed in the normalized space remain valid in physically meaningful units.

*Table 12.* Zero-shot prediction errors in the original physical space. We report MSE on representative state dimensions from three robotic systems. Lower is better.

| Method | Walker root velocity (m/s) | Walker foot angle (rad) | Hopper leg angle (rad) | Hopper foot angle (rad) | Franka $y$ position (m) | Franka pitch (rad) |
|---|---|---|---|---|---|---|
| | MSE ↓ | MSE ↓ | MSE ↓ | MSE ↓ | MSE ↓ | MSE ↓ |
| MLPEnsemble | 4.623 | 1.339 | 0.089 | 0.685 | 0.068 | 0.547 |
| TDM | 4.654 | 0.615 | 0.134 | 0.372 | 0.653 | 0.390 |
| TrajWorld | 3.202 | 0.779 | 0.093 | 0.468 | 0.538 | 0.162 |
| Ours | **2.544** | **0.183** | **0.045** | **0.244** | **0.039** | **0.130** |

### E.2. Impact of Pretraining on Few-shot Learning

To further quantify the impact of pretraining, we compare few-shot learning curves of `WestWorld` with and without pretraining. As illustrated in Fig. 6, the pretrained model achieves substantially lower MSE across all 10 fine-tuning episodes on three robots. The performance gap remains large even after convergence, indicating that pretrained `WestWorld` not only accelerates adaptation but also leads to better final accuracy. These results confirm that our system-aware pretraining enables the model to capture distinct robotic dynamics priors that can be efficiently adapted to new systems with limited data.

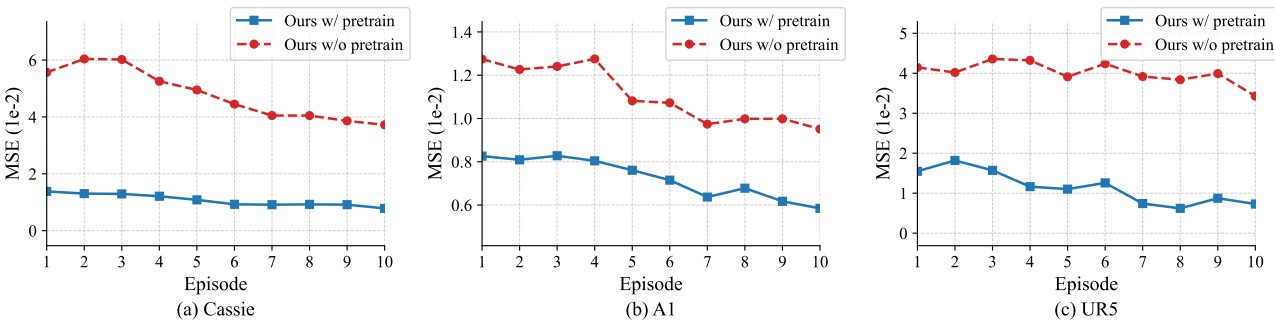

*Figure 6.* The effect of pre-training on few-shot learning for three different robotic systems: (a) Cassie, (b) A1 and (c) UR5 Robots.

## F. Discussion

**Efficient Parameter Finetuning.** When employing the pre-trained model to downstream tasks, fully fine-tuning all model parameters can be memory-intensive. Motivated by parameter-efficient finetuning, we study whether few-shot adaptation can be achieved by updating only a small subset of parameters. In our experiments, we freeze the backbone and fine-tune only (i) the Sys-MoE layers, (ii) the learnable query tokens, and (iii) the final decoder linear head.

Table 13 shows the experimental results by fine-tuning only the last $j$ Sys-MoE layers ($j \in \{2, 4, 6\}$), and compares them with the full fine-tuning method. It can be seen from this table that the increase of $j$ can improve prediction performance across Cassie, A1, and UR5 systems. However, fine-tuning only the last few Sys-MoE layers still achieve high accuracy compared to the baseline. Specifically, fine-tuning the last two Sys-MoE layers updates only 21.91% of the parameters, yet it outperforms the fully fine-tuned TrajWorld baseline on all three systems. This result implies that parameter-efficient updates to the Sys-MoE components remain effective even under domain shifts.

*Table 13.* We compare fine-tuning the last 2/4/6 Sys-MoE layers against full fine-tuning and the best-performing SOTA baseline (TrajWorld). All experiments use a fixed random seed. Lower is better.

| Method | Tuned Layers | Tuned Params | MSE ($\times 10^{-2}$) $\downarrow$ | | |
|---|---|---|---|---|---|
| | | | Cassie | A1 | UR5 |
| Trajworld | Full | 10.0 M | 1.674 | 0.931 | 2.616 |
| Ours | Last 2 Sys-MoE | 3.2 M (21.91%) | 1.249 | 0.775 | 1.052 |
| | Last 4 Sys-MoE | 6.3 M (43.37%) | 1.074 | 0.762 | 1.031 |
| | Last 6 Sys-MoE | 9.5 M (64.83%) | 1.055 | 0.744 | 0.895 |
| | Full | 14.6 M (100%) | **0.785** | **0.636** | **0.761** |

**Computational Cost.** We further compare the inference-time latency of our method with SOTA autoregressive transformer baselines, namely *TDM* (Schubert et al., 2023) and *TrajWorld* (Yin et al., 2025). All methods use a fixed-length history window of $T_{\text{hist}} = 50$ steps as input, with batch size $B = 4$, observation dimension $D_o = 78$, and action dimension $D_a = 21$. We evaluate multi-step prediction horizons $H \in \{10, 30, 50, 70, 100\}$ on a single NVIDIA RTX A6000 GPU. We report the mean and standard deviation of wall-clock latency (in milliseconds) over 10 repeated runs after warm-up, with CUDA synchronization enabled. For autoregressive baselines, we follow the setup in (Yin et al., 2025) and use KV-cache decoding to reduce per-step computational overhead. Table 14 summarizes the inference latency for different methods. Our approach consistently achieves fastest inference for long-horizon predictions except for $H$=10. This advantage arises from our use of learnable query embeddings, which enable the model to predict all $H$ future steps in a single forward pass. As a result, our method is approximately $3.09\times$ faster than *TrajWorld* and $10.73\times$ faster than *TDM* when $H = 100$.

*Table 14.* Wall-clock latency (ms, mean$\pm$std) for multi-step prediction on a single NVIDIA RTX A6000. All methods take $T_{\text{hist}} = 50$ history steps as input ($B = 4$, $D_o = 78$, $D_a = 21$). Results are averaged over 10 repeated runs after warm-up. Lower is better.

| Method | Latency (ms) $\downarrow$ | | | | |
|---|---|---|---|---|---|
| | $H = 10$ | $H = 30$ | $H = 50$ | $H = 70$ | $H = 100$ |
| TDM | 388.85$\pm$0.41 | 697.89$\pm$0.64 | 1099.63$\pm$1.41 | 1602.34$\pm$1.33 | 2558.94$\pm$1.64 |
| TrajWorld | **94.62$\pm$0.30** | 236.63$\pm$0.33 | 379.43$\pm$0.59 | 522.37$\pm$0.43 | 736.67$\pm$0.91 |
| Ours | 113.61$\pm$0.71 | **138.03$\pm$1.45** | **159.90$\pm$0.97** | **183.96$\pm$1.01** | **238.52$\pm$0.47** |

# G. Real-world Deployment

This section provides the implementation details for the real-world Unitree Go1 deployment. To enable real-time control at a high control rate (90 Hz), we distill the pretrained world models into lightweight student models via knowledge distillation (KD) (Hinton et al., 2015), and then fine-tune them using simulation data collected from the downstream Go1 control task. For a fair comparison, we apply the same distillation, fine-tuning, and MPPI deployment protocol to both `WestWorld` and the strongest baseline `TrajWorld`.

**Distillation objective.** For each teacher model, we construct a lightweight student by reducing the backbone depth to two blocks while keeping the same discretized prediction head over $K$ uniform bins as in Eq. (10). Let $\boldsymbol{P}_\ell^{(m)}$ denote the categorical prediction over $K$ bins produced by the pretrained teacher model for channel $m$ at token index $\ell$, and let $\widetilde{\boldsymbol{P}}_\ell^{(m)}$ be the corresponding student prediction. We train the student with a convex combination of the original next-token cross-entropy loss and a soft distillation loss:

$$\mathcal{L}_{\mathrm{KD}} = \alpha\,\mathcal{L}_{\mathrm{CE}} + (1-\alpha)\,\mathcal{L}_{\mathrm{soft}}, \tag{37}$$

where $\mathcal{L}_{\mathrm{CE}}$ is the same next-token loss in Eq. (11), and the soft loss matches the student distribution to the teacher distribution:

$$\mathcal{L}_{\mathrm{soft}} = -\sum_{\ell=1}^{L-1} \sum_{m=1}^{M_s} \boldsymbol{P}_\ell^{(m)\top} \log \widetilde{\boldsymbol{P}}_\ell^{(m)}. \tag{38}$$

We set $\alpha = 0.9$ in the KD experiments.

**Distillation and fine-tuning settings.** For both `WestWorld` and `TrajWorld`, we first distill the student model on the same pretraining trajectories by using the corresponding teacher outputs $\boldsymbol{P}_\ell^{(m)}$ as soft targets, while keeping the ground-truth token supervision through $\mathcal{L}_{\mathrm{CE}}$. After distillation, both students are fine-tuned on the same simulation dataset collected for downstream Go1 control (Section C.5), following the same optimizer and early-stopping protocol as in the downstream control training settings (Section D.3). During real-world deployment, both student models are used as the dynamics predictor inside the same MPPI controller with identical planning hyperparameters.

