# OpenReview forum: "WestWorld: A Knowledge-Encoded Scalable Trajectory World Model for Diverse Robotic Systems"
_ICML.cc/2026/Conference — ICML 2026 spotlight_

### Official Review · Reviewer_614q · 2026-02-16

**Soundness:** 2
**Presentation:** 3
**Significance:** 2
**Originality:** 3
**Overall Recommendation:** 4
**Confidence:** 3

**Summary:**

A core problem considered by the manuscript is how to scale a trajectory world model across diverse robot morphologies while maintaining zero-/few-shot generalization. The paper proposes WestWorld, combining (i) a system-aware Mixture-of-Experts (Sys-MoE) that routes experts via a learnable system embedding to mitigate inter-robot interference, and (ii) morphology-aware structural/channel embeddings that inject physical connectivity as inductive bias. The paper’s important finding concerns consistent gains in zero-/few-shot trajectory prediction and downstream model-based control, including deployment on a physical Unitree Go1 through distillation and fine-tuning.

**Compliance With Llm Reviewing Policy:**

Affirmed.

**Final Justification:**

My concerns have been addressed. Although the real-world experiments are quite limited, the paper still makes a meaningful contribution. Therefore, I will raise my score to a positive evaluation and encourage the authors to further improve the work.

**Key Questions For Authors:**

See weaknesses.

**Limitations:**

yes

**Strengths And Weaknesses:**

# Strengths
1. The design is well-motivated (task interference + missing morphology bias) and is validated with targeted ablations showing both Sys-MoE and knowledge-encoded structural embedding are necessary for generalization.
2. The paper states clear challenges and contributions and provides a structured experimental story (scalability/generalization, ablations, control/deployment).

# Weaknesses
1. Core components resemble established recipes (MoE routing + morphology encoding); the combination is reasonable but not conceptually surprising, and the novelty is more “engineering integration” than new principle.
2. The current model focuses on trajectory-only modeling (no visual observations), limiting applicability to vision-conditioned robotics without additional modeling work.
3. Real-world validation is limited and insufficiently thorough. Only a small-scale Go1 deployment demo is presented, without quantitative comparison against baselines in the real setting. The lack of controlled, side-by-side evaluation significantly weakens the empirical support for real-world robustness and superiority.

---

> ### Author Rebuttal · Authors · 2026-03-29
>
> **Q4.1: Core components resemble established recipes (MoE routing + morphology encoding); the combination is reasonable but not conceptually surprising, and the novelty is more “engineering integration” than new principle.**
>
> **A4.1:** We would like to highlight the main contributions of our work as follows (Reviewers LCY8 and znwe also mentioned them).
> - Our Sys-MoE is not just an engineering integration. In contrast, we design this component based on the priciple that different robotic systems are governed by various system dynamics. This inspires us to develop a system-aware MoE with continuous-time state space models (SSM) that approximate the underlying governing equations by routing different experts.
> - We are the first to integrate morphology encoding into world models, which introduce inductive bias to enhance model generalization to unseen systems.
>
> **Q4.2: The current model focuses on trajectory-only modeling (no visual observations), limiting applicability to vision-conditioned robotics without additional modeling work.**
>
> **A4.2:** We have discussed the limitation in Sec. 6. We will explore it for future work.
>
> **Q4.3: Real-world validation is limited and insufficiently thorough. Only a small-scale Go1 deployment demo is presented, without quantitative comparison against baselines in the real setting. The lack of controlled, side-by-side evaluation significantly weakens the empirical support for real-world robustness and superiority.**
>
>
> **A4.3:** Per your suggestion, we conducted an additional real-world comparison against the strongest baseline, TrajWorld, given the limited rebuttal period. We present the quantitative results in Table 1 and provide a side-by-side video comparison on our demo site. (Please refer the link in the abstract)
>
> The results show that our method successfully completes the real-world Go1 straight-walking task and achieves a high reward, whereas TrajWorld fails to enable the robot to reliably stand up and walk toward the target. This failure mode is consistent with our simulation results, where TrajWorld already failes on the Go1 walk straight task before distillation. This gap becomes more pronounced in real-world deployment because the sim-to-real domain shift further amplifies Trajworld's prediction errors. Since MPPI relies on accurate long-horizon rollouts to evaluate sampled action sequences, such prediction errors can lead to poor action selection and control failure. In contrast, our method remains sufficiently accurate to support stable control in the real setting.
>
> **Table 1.** Real-world Go1 straight-walking performance. We report the accumulated reward achieved during deployment.
> | Method | Accumulated reward ↑ |
> |---|---:|
> | TrajWorld | -5.25e-18 |
> | WestWorld (ours) | **1.2521** |

---

> > ### Author Rebuttal · Reviewer_614q · 2026-04-02
> >
> > My concerns have been addressed. Although the real-world experiments are quite limited, the paper still makes a meaningful contribution. Therefore, I will raise my score to a positive evaluation and encourage the authors to further improve the work.

---

> > > ### Author Response · Authors · 2026-04-02
> > >
> > > Dear Reviewer,
> > >
> > > Thank you for raising your score. We will incorporate your comments in our revised version. We really appreciate your suggestions to improve our work.

---

### Official Review · Reviewer_znwe · 2026-03-10

**Soundness:** 3
**Presentation:** 4
**Significance:** 3
**Originality:** 3
**Overall Recommendation:** 4
**Confidence:** 4

**Summary:**

This paper proposes WestWorld, a scalable trajectory world model designed to learn dynamics across diverse robotic systems. The authors argue that existing trajectory world models struggle to scale to heterogeneous robots due to parameter interference and the lack of physical inductive bias from robot morphology. To address these issues, the paper introduces two key components: (1) a system-aware mixture-of-experts architecture (Sys-MoE) that dynamically routes inputs to specialized experts using a learnable system embedding, and (2) a morphology-aware structural embedding that aligns trajectory representations with robot physical structures to improve generalization. The results show that WestWorld consistently outperforms baselines such as MLP Ensemble, TDM, and TrajWorld in long-horizon trajectory prediction and improves downstream control performance.

**Compliance With Llm Reviewing Policy:**

Affirmed.

**Final Justification:**

During the rebuttal, the authors addressed my concern. I will maintain my positive score.

**Key Questions For Authors:**

In particular, it would be important to include comparisons such as a dense baseline model versus the MoE architecture, MoE without morphology embedding, and different routing strategies within the MoE framework. These controlled ablation experiments would help verify whether both components independently contribute to the observed performance improvements and clarify the relative impact of each design choice. Without such analysis, it remains difficult to determine which component primarily drives the reported gains.

**Limitations:**

The above has already explained my question.

**Strengths And Weaknesses:**

# strength
The system-aware MoE architecture is a reasonable design for scaling dynamics learning. The proposed Sys-MoE routing mechanism allows the model to dynamically combine different experts depending on the robot system. Incorporation of morphology knowledge provides meaningful inductive bias. Introducing structure-based channel embeddings to encode robot morphology is a valuable idea. Robot dynamics strongly depend on morphology, and explicitly incorporating structural information is a principled way to improve generalization across robots.

# Weakness
Although the proposed model introduces two key innovations, the paper does not provide sufficiently detailed ablation experiments to clearly isolate the contribution of each component.

---

> ### Author Rebuttal · Authors · 2026-03-29
>
> **Q3.1: Although the proposed model introduces two key innovations, the paper does not provide sufficiently detailed ablation experiments to clearly isolate the contribution of each component.**
>
> **A3.1:** We thank the reviewer for this insightful suggestion regarding the need for clearer component-wise ablations. We respectfully note that the requested ablation study was already included in the original submission (Table 4, Section 5.2), where we examined the contribution of the two core components: **Sys-MoE** and **structural embedding**. For the reviewers’ convenience, we reproduce the results below.
>
> From this **Table 1**, we can make two observations. First, replacing Sys-MoE with a parameter-matched dense SSM degrades performance on all three systems, which suggests that the gain does not come merely from model size. Instead, the Sys-MoE design is important for jointly modeling diverse robotic dynamics and for reducing inter-task interference during large-scale pretraining.
>
> Second, removing the structural embedding also causes a clear performance drop, especially on Walker2D and Hopper, which involve more complex morphology. This supports our claim that structural embedding provides a useful morphology-aware inductive bias, helping the model align trajectory representations with physical structure and improving zero-shot generalization to unseen robots. Overall, these results show that the two proposed components essential for model generalization and scalable pretraining.
>
> **Table 1:** Ablation results under the zero-shot setting. We ablate the Sys-MoE layer by replacing it with a dense SSM, and ablate the structural encoding by removing it during pretraining. Results are computed in the normalized space and reported as MAE and MSE.
> | Method | Sys-MoE | Structural embedding | Walker2D MAE ($\times 10^{-2}$) ↓ | Walker2D MSE ($\times 10^{-2}$) ↓ | Hopper MAE ($\times 10^{-2}$) ↓ | Hopper MSE ($\times 10^{-2}$) ↓ | Franka MAE ($\times 10^{-2}$) ↓ | Franka MSE ($\times 10^{-2}$) ↓ |
> |---|---|---|---:|---:|---:|---:|---:|---:|
> | Ours w/o Sys-MoE | ✗ | ✓ | 18.707 | 6.797 | 15.978 | 4.630 | 9.392 | 2.873 |
> | Ours w/o Structural embedding | ✓ | ✗ | 21.156 | 7.872 | 16.227 | 4.990 | 7.897 | 2.707 |
> | Ours | ✓ | ✓ | **16.350** | **5.064** | **13.731** | **3.368** | **7.737** | **2.539** |
>
> **Q3.2: In particular, it would be important to include comparisons such as a dense baseline model versus the MoE architecture, MoE without morphology embedding, and different routing strategies within the MoE framework. These controlled ablation experiments would help verify whether both components independently contribute to the observed performance improvements and clarify the relative impact of each design choice.**
>
> **A3.2:** We address this point in our response to **A3.1** above.

---

### Official Review · Reviewer_LtV6 · 2026-03-13

**Soundness:** 2
**Presentation:** 2
**Significance:** 2
**Originality:** 3
**Overall Recommendation:** 4
**Confidence:** 4

**Summary:**

This paper introduce the WestWorld, which is a low-level trajectory world model that has two main features:
1. a knowledge-encoded structural embeddings for improved generalization to unseen robot dynamics
2. the system-aware mixture-of-expert layer with each expert specialized in a set of basic dynamics
The proposed method is evaluated in both simulation environments and real-world robot systems with zero-shot and few-shot setting and is compared against some other baseline methods. The results show the effectiveness of the proposed method The ablation studies validate the effectiveness of both introduced modules.

**Compliance With Llm Reviewing Policy:**

Affirmed.

**Final Justification:**

The rebuttal has addressed most of my concerns. That's said, there are still room for further improvements.

**Key Questions For Authors:**

I've raised many questions in the weakness part, some clear explaination address those confusion and concerns will be greatly appreciated. In addition, the caption and plots in figure 2 is mismatched. The appendix parts fail to include all of the experimental details. For example, where are the settings / rollouts samples of Cassie experiments?

**Limitations:**

Yes

**Strengths And Weaknesses:**

Strength
1. The idea of injecting knowledge-encoded embedding and system-aware expert dynamics into the world model prediction is clear and effective.
2. The ablation study is clear and supportive for the effectiveness of the two proposed modules.

Weakness
1. The manuscript does not provide a clear explanation on how the system-aware MoE work, specially on how the experts are trained. In line 207, "each expert specializes in a basis dynamics module", what is a basis dyanmics module? How to select these basis dynamics module? How the experts are trained?
2. The proposed model is pre-trained on a large dataset consists of some similar environments such as TD-MPC2, DB-1 and Modular-RL. In TD-MPC2 and DB-1 there are Hoppers; in ExORL, RL Unplugged, DB-1, TD-MPC2, DB-1 and Modular-RL there are Walker2Ds. Given the low-level world model setting with pure state input (no rgb images), I don't understand why Walker2d and Hopper can be classified as zero-shot environments. If each expert is trained on each type of environment, then a MoE style WM will for sure surpass a general WM on these tasks with proper routing.
3. According to Figure 2, MLPEnsemble, TDM, and Trajworld are almost identically bad on dyanmic prediction on Walker and Hopper, but why in table 3 Trajworld has almost doubled performance in comparison?
4. I'm not quite convinced by the real-world G1 experiments. The world model used for this MPPI task is distilled from WestWorld and then further finetuned with simulation data. As a simple locomotion task with fixed gait pattern, how to tell whether it's not by overfitting?
5. The overall delivery of this paper is not ideal, a more clear narrative and structure will be appreciated.

---

> ### Author Rebuttal · Authors · 2026-03-29
>
> **Q2.1: The Sys-MoE design explanation is unclear; What does “basis dynamics module” mean, How to select these basis dynamics module? How the experts are trained?**
>
> **A 2.1:**
> The high-level idea of our system-aware MoE is to approximate the underlying system dynamics by routing inputs to different experts via a learnable system embedding. Concretely, after obtaining latent states, we concat a learnable system embedding and feed the combined states into an SSM. The output corresponding to the system embedding is then used by a router to compute soft mixture weights over all experts. Each expert processes the hidden states, and the final representations are obtained as a weighted sum of the expert outputs.
>
> Note that, the experts are not predefined basis dynamics and are not manually selected. Instead, each expert is an MLP that learns a different latent component of the dynamics space. All experts together with the router and the SSM are trained jointly in an end-to-end manner. We will revise the sentence as follows: "in which each expert tries to learn part of underlying system dynamics" to avoid confusion.
>
> **Q2.2: Why are Walker and Hopper considered zero-shot if similar systems appear in pretraining? If each expert is trained on each type of environment, then a MoE WM will for sure surpass a general WM with proper routing.**
>
> **A2.2:** In our setting, the zero-shot Hopper and Walker2d evaluations are conducted on the D4RL data excluded from pretraining. More importantly, the pretraining tasks and zero-shot test tasks are based on different environments and system dynamics. Our pretraining uses **DeepMind Control Suite (DMC)** tasks, whereas the test data come from **D4RL / OpenAI Gym** environments. The DMC and D4RL versions are defined by different MuJoCo XML models and therefore correspond to different dynamical systems. For example, DMC Hopper includes an additional waist joint / DOF and uses 4 actuators, whereas D4RL Hopper uses only 3 actuators. DMC Walker differs from D4RL Walker2d in leg displacement, hip joint range, and actuator gear settings.
>
> Please also note that we do not train one expert per environment type. All experts are trained jointly in a single pretraining stage over diverse systems, and Sys-MoE routes inputs by automatically activating suitable expert combinations.
>
>
> **Q2.3: Explain why Figure 2 shows similarly poor zero-shot results for all methods, but Table 3 shows TrajWorld performing much better.**
>
> **A2.3:** Figure 2 reports **zero-shot** prediction, whereas Table 3 reports downstream control with task-specific **finetuning**. In zero-shot, all baselines perform poorly due to task interference during large-scale pretraining and the lack of morphology-aware inductive bias. After finetuning, TrajWorld benefits from its temporal-variate attention design, which captures spatiotemporal dependencies better than TDM and MLPEnsemble. A similar trend appears in the few-shot results in Table 2.
>
> **Q2.4: About the real-world Go1 result may reflect overfitting, since the WM is distilled and further fine-tuned on simulation data**
>
> **A2.4:** Our real-world Go1 result is unlikely to be explained by simple overfitting for two reasons. First, our control setting places a challenging requirement on world-model generalization. The controller performs **sample-based MPPI**, where the world model must accurately evaluate a large number of **randomly sampled action sequences** over a long horizon. The final control action is obtained by weighting these predicted trajectories according to their estimated returns. If the model merely overfits to a narrow fixed gait pattern, it would produce inaccurate long-horizon rollouts for off-pattern sampled actions, leading to incorrect trajectory weighting and, consequently, poor action selection.
>
> Second, the model is **trained only in simulation** but evaluated in **real-world deployment**, so it must generalize across a clear **sim-to-real domain gap**. A model that is simply overfitted to the training distribution would typically suffer from even larger rollout errors under this shift, which would further degrade MPPI performance in real-world deployment.
>
> **Q2.5: About the paper would benefit from a clearer narrative and structure.**
> **A 2.5:** We will revise the narrative and structure in our version.
>
> **Q2.6: (1) Figure 2 has a caption/plot mismatch. (2) missing some experimental details, e.g., the Cassie settings**
>
> **A2.6:** (1)Regarding Figure 2, we will correct the mismatch between the caption and the plot order in the revision.
>
> (2)The detailed dataset descriptions and experimental settings are already included in the appendix, specifically in **Appendix C** and **Appendix D**. For the Cassie experiments in particular, the dataset setup is provided in **Appendix C.3 (Few-shot experiment datasets)**, and the corresponding training and evaluation details are provided in **Appendix D.2**, under **Few-shot evaluation settings**.

---

> > ### Author Rebuttal · Reviewer_LtV6 · 2026-04-04
> >
> > I will raise my score to a positive evaluation

---

### Official Review · Reviewer_LCY8 · 2026-03-13

**Soundness:** 4
**Presentation:** 4
**Significance:** 3
**Originality:** 3
**Overall Recommendation:** 5
**Confidence:** 3

**Summary:**

The paper tackles the problem of building a unified trajectory world model across diverse robotic morphologies. The proposed WestWorl leverages a System-aware Mixture-of-Experts architecture and morphological structural embeddings via LCRS trees to scale across different dynamics while maintaining zero-shot generalizability. The motivation is clear, and the experimental evaluation is extensive, spanning zero-shot generalization, few-shot adaptation, and downstream model-based control tasks.

**Compliance With Llm Reviewing Policy:**

Affirmed.

**Final Justification:**

This paper studies scalable trajectory world modeling across diverse robotic systems. In my assessment, the work is strong in terms of soundness and clarity. The motivation is clear, the method is reasonably presented, and the experimental evaluation is broad. The paper evaluates the method on zero-shot prediction, few-shot adaptation, scalability, and downstream model-based control, which together provide a solid basis for assessment.

My initial concerns were mainly about the empirical evidence and presentation: the possible precision loss caused by discretization, the lack of metrics reported in physical units, the limited scalability curve relative to the full pretraining scale, and the limited diversity of the real-world zero-shot evaluation. After reading the rebuttal, I believe the authors addressed these concerns effectively.

My final recommendation is based mainly on the overall technical solidity of the work, the reasonableness of the method, the strength of the empirical validation, and whether the rebuttal addressed my main concerns. From that perspective, the rebuttal increased my confidence in the paper and changed my evaluation positively.

Taking both the paper and the rebuttal into account, I believe the work meets the standard for Accept. The rebuttal addressed my main concerns in a substantive way, and I accordingly raised my score.

**Key Questions For Authors:**

See Weakness.

**Limitations:**

Yes,  the authors adequately discussed the limitations and potential negative societal impact of their work

**Strengths And Weaknesses:**

### Strengths

* **Innovative Architecture:** The integration of structural priors via LCRS trees and dynamic routing through the Sys-MoE provides a principled approach to mitigating negative transfer and gradient conflicts during multi-robot joint training.


* **Comprehensive Evaluation Setup:** The experimental design covers a widerange of settings, effectively testing the model in simulation and real-world environments, as well as its utility as a dynamics predictor for downstream MPPI control. The experiment of this paper is solid, thorough, and strong.


### Weaknesses & Questions

**1. Tokenization and Precision Loss**
The method discretizes state and action into 256 bins. While this facilitates next-token prediction, it inherently introduces inherent precision loss for the model, which can be highly detrimental to fine-grained downstream control tasks.

* *Question:* I would like the authors to discuss the quantitative impact of this precision loss on model reconstruction accuracy. Furthermore, please compare and contrast this uniform binning tokenization with other representation learning methods (e.g., FAST Tokenizer, VQ-VAE), which might preserve continuous semantics and structural information more effectively.

**2. Evaluation Metrics in Physical Space**
Throughout the experiments, the MSE and MAE metrics are reported exclusively in the normalized space. Reporting errors solely in the normalized space is not aligned with the actual physical meaning of the results.

* *Question:* Could the authors additionally report the unnormalized metrics in their true physical spaces (e.g., radians, meters/second)? This would provide a much more intuitive understanding of the model's actual predictive accuracy.

**3. Completeness of the Scalability Experiments**
In the scalability evaluation (Figure 4), the x-axis limits the number of environments to a maximum of 30. However, the paper states that the pretraining phase utilizes 89 environments (80 simulated + 9 real-world).

* *Question:* To fully substantiate the claim that Sys-MoE effectively scales without performance degradation, why not extend the error curves to encompass a larger number of environments (e.g., up to the full 89)? I highly recommend reporting the performance curve across more environments to demonstrate true scalability.


**4. Diversity in Zero-Shot Generalization**
The zero-shot generalization evaluation only evaluated Franka as the real-world robotic arm.

* *Question:* To strengthen the core claim of broad generalizability to unseen robotic systems, could the authors consider adding zero-shot evaluations on other physical robot arms (e.g., UR5e) or platforms with distinct morphologies?

---

> ### Author Rebuttal · Authors · 2026-03-29
>
> **Q 1.1: (1) discuss the quantitative impact of precision loss on model reconstruction accuracy. (2) Compare this uniform binning method with other representation learning methods (e.g., FAST Tokenizer, VQ-VAE).**
>
> **A 1.1:**
> (1) Our tokenization follows prior works [1,2] that discretizes continuous targets into bins for more stable training. Per your advice, we add a quantitative ablation on the number of bins (128/256/512) and report the prediction errors in Table 1 below. Empirically, the error with 256 bins is comparable to that with 512 bins, while incurring lower computational cost. We thus choose 256 as a practical trade-off between precision and efficiency.
>
> (2) Regarding alternatives, FAST Tokenizer is mainly designed for **chunk-wise** action tokenization, whereas our model needs to learn **step-wise** state-action transition dynamics, making chunk-wise tokenization less suitable. VQ-VAE is also less straightforward here, as it requires additional training and is harder to apply across robotic systems with heterogeneous dimensions. In contrast, our tokenization is non-parametric and easy to scale across diverse robots.
>
> **Table 1**: Quantitative impact of the number of bins on the Walker system. Errors are reported as MAE and MSE ($\times 10^{-2}$).
>
> | Method | Walker |  |
> |---|---:|---:|
> |  | MAE↓ | MSE↓ |
> | 128 bins | 1.455 | 0.098 |
> | 256 bins | 1.415 | 0.095 |
> | 512 bins | 1.404 | 0.094 |
>
> >[1] TrajWorld, ICML 2025.
> >
> >[2] Stop Regressing. ICML, 2024.
>
> **Q 1.2: Report the unnormalized metrics in their true physical spaces (e.g., radians, meters/second).**
>
> **A 1.2:** We additionally report the unnormalized zero-shot errors in the physical space (e.g., radians, m/s) in Table 2. Our method still achieves the lowest errors on all three systems, showing that the performance gains remain consistent in physically meaningful units. We will include the full results in our revision.
>
> **Table 2.** Zero-shot performance of different models on three dynamical systems. Errors are computed in the physical space and reported as MSE.
>
> | Method | Walker(rootx vellocity) | Walker(foot angle) | Hopper(leg angle) | Hopper(foot angle) | Franka(y pose) | Franka(pitch)|
> |---|---:|---:|---:|---:|---:|---:|
> |  | MSE↓ | MSE↓ | MSE↓ | MSE↓ | MSE↓ | MSE↓ |
> | MLPEnsemble | 4.623 | 1.339 | 0.089 | 0.685 | 0.068 | 0.547 |
> | TDM | 4.654 | 0.615 | 0.134 | 0.372 | 0.653 | 0.390 |
> | Trajworld | 3.202 | 0.779 | 0.093 | 0.468 | 0.538 | 0.162 |
> | **Ours** | **2.544** | **0.183** | **0.045** | **0.244** | **0.039** | **0.130** |
>
>
> **Q 1.3: Extend the error curves to encompass a larger number of environments (e.g., up to the full 89)?**
>
> **A 1.3:** Per your advice, we conducted additional scaling experiments by further increasing the number of pretraining environments from **30 to 50, 60, and 89**. The updated results are reported in **Table 3**.
>
> These additional results further strengthen our scalability claim. As the number of environments increases, our method remains relatively stable, with test MAE rising only moderately. For example, when the number of environments increases from 1 to 20, our MAE changes from 1.814 to 2.587 (×10^-2). In contrast, TrajWorld degrades much more substantially over the same range, with MAE increasing from 3.038 to 7.996 (×10^-2).
>
> **Table 3.** Comparison between our method against the best performing SOTA by scaling the number of environments. We report test MAE ($\times 10^{-2}$).
>
> | Method | k=1 | k=10 | k=20 | k=30 | k=50 | k=60 | k=89 |
> |---|---:|---:|---:|---:|---:|---:|---:|
> | TrajWorld | 3.038 | 6.992 | 7.996 | 8.088 | 8.820 | 9.806 | 11.788 |
> | Ours | **1.814** | **2.549** | **2.587** | **2.683** | **3.391** | **3.578** | **3.790** |
>
> **Q 1.4: Add zero-shot evaluations on other physical robot arms (e.g., UR5e) or platforms with distinct morphologies.**
>
> **A 1.4:** We additionally evaluated our method on two more real-world robotic systems, **Unitree A1**[1] and **UR5**[2], which have substantially different morphologies from the pretraining data. As shown in Table 4, WestWorld consistently achieves the best performance among all baselines.
>
> This result further supports our claim of broad generalizability to unseen robotic platforms. We attribute this to the system-aware MoE architecture and the morphology-aware structural inductive bias, which together help WestWorld route appropriate experts and better approximate unseen dynamics under domain shift.
>
> **Table 4**: Zero-shot performance of all models on A1 and UR5.
> | Method | A1 (×10^-2) MAE↓ | A1 (×10^-2) MSE↓ | UR5 (×10^-2) MAE↓ | UR5 (×10^-2) MSE↓ |
> |---|---:|---:|---:|---:|
> | MLPEnsemble | 13.123 | 3.252 | 14.805 | 4.803 |
> | TDM | 7.379 | 1.228 | 19.917 | 6.182 |
> | Trajworld | 5.917 | 1.262 | 9.428 | 2.812 |
> | **Ours** | **5.433** | **0.866** | **9.332** | **2.164** |
>
> >[1] SayTap: Language to Quadrupedal Locomotion. CoRL 2023.
> >
> >[2] Learning modular language-conditioned robot policies through attention. Autonomous Robots 2023.

---

> > ### Author Rebuttal · Reviewer_LCY8 · 2026-04-02
> >
> > Thank you for the additional results provided in the rebuttal. These results have significantly strengthened the paper, and my main concerns have been adequately addressed. Based on the strong empirical evidence now provided, I would like to raise my score to 5 (Accept). I would also recommend that these results be included in the main manuscript.

---

> > > ### Author Response · Authors · 2026-04-02
> > >
> > > Dear Reviewer,
> > >
> > > Thank you for raising the score to 5. We will incorporate your comments in our revised version. By the way, could you please modify the score in the **Overall Recommendation** since it still maintains 4. Thank you very much for your support!

---

### Decision · Program_Chairs · 2026-04-30

**Decision:**

Accept (spotlight)

**Comment:**

The paper tackles the problem of building a scalable, unified trajectory world model across diverse robotic morphologies, aimed at capturing dynamics across different systems while maintaining zero-/few-shot generalization. The initial reviews raised several concerns, including, for example, the quantitative impact of precision loss, evaluation metrics, scalability evaluation, generalization, confusion and issues in the explanations, the need for detailed ablation studies, and limitations in the real-world experiments. However, these concerns were well addressed in the rebuttal, and as a result, three reviewers raised their scores. The final ratings were clearly positive, with one accept and three weak accepts. Based on this, the AC recommends acceptance of this paper. It is strongly encouraged that the final version include the responses and additional experimental results provided during the rebuttal phase.